# Accurately clustering biological sequences in linear time by relatedness sorting

Erik Wright ● [1,2] ✉

Clustering biological sequences into similar groups is an increasingly important task as the number of available sequences continues to grow exponentially. Search-based approaches to clustering scale super-linearly with the number of input sequences, making it impractical to cluster very large sets of sequences. Approaches to clustering sequences in linear time currently lack the accuracy of super-linear approaches. Here, I set out to develop and characterize a strategy for clustering with linear time complexity that retains the accuracy of less scalable approaches. The resulting algorithm, named Clusterize, sorts sequences by relatedness to linearize the clustering problem. Clusterize produces clusters with accuracy rivaling popular programs (CD-HIT, MMseqs2, and UCLUST) but exhibits linear asymptotic scalability. Clusterize generates higher accuracy and oftentimes much larger clusters than Linclust, a fast linear time clustering algorithm. I demonstrate the utility of Clusterize by accurately solving different clustering problems involving millions of nucleotide or protein sequences.

There are a myriad of applications for clustering biological sequences into groups based on their similarity[1]. For example, clustering is often used to reduce redundancy among sequences, bin metagenomic or transcriptomic sequencing reads, cluster protein sequences into homologous groups, or define operational taxonomic units (OTUs) among phylogenetic marker gene sequences[2]. This wide variety of applications has resulted in a plethora of clustering algorithms and similarity measures. Nevertheless, all modern clustering programs are expected to scale to continually increasing numbers of sequences. It is now common to have millions of sequences to cluster. This necessitates algorithmic approaches that can maintain accuracy while scaling in linear time with the number of input sequences.

Modest numbers of sequences can be clustered with exact hierarchical algorithms that exhaustively calculate pairwise distances and guarantee linkage (i.e., average, complete, or single) among all clustered sequences[3,4]. Larger sets of sequences require inexact clustering approaches that only establish linkage to one representative sequence in each cluster. The objective of inexact clustering is deceptively simple: for each input sequence, check whether it is similar enough to any of the existing cluster representatives and, if not, form a new cluster. However, it is possible to see that this formulation of clustering has

quadratic time (i.e., $O(N^2)$) complexity in the worst case. The worst case occurs when most sequences are the only member of their cluster, which is likely to happen when the input sequences are very diverse or as the similarity threshold approaches 100%. There are many applications of clustering where the goal is to cluster at high similarity thresholds, such as dereplication and finding nearly identical sequences across samples. Hence, a reduction in complexity is required in order for clustering to scale to large numbers of input sequences, ideally with negligible loss in accuracy.

Popular heuristic algorithms, including CD-HIT[5], MMseqs2[6], and UCLUST[7], iteratively add sequences to a growing list of cluster representatives if no hits are found in the existing set. Since the number of cluster representatives ($M$) tends to be proportional to the number of input sequences ($N$), these approaches often scale super-linearly (i.e., $O(M*N)$) even with the use of heuristics to accelerate distance calculations[6,8–10]. A related strategy with similar scalability is to search for each new cluster representative within the remaining (unassigned) sequences[11–13], thereby recruiting new sequences to each cluster. Achieving quasi-linear (i.e., $O(N*log(N))$) scalability is possible with methods that divide the input sequences into separate sets before clustering[14,15]. Linear time ($O(N)$) clustering was achieved by Linclust

[1]Department of Biomedical Informatics, University of Pittsburgh, Pittsburgh, PA, USA. [2]Center for Evolutionary Biology and Medicine, Pittsburgh, PA, USA.
✉e-mail: eswright@pitt.edu

using k-mer grouping prior to clustering, although Linclust was less sensitive than super-linear time algorithms[16].

Here, I set out to develop a new clustering algorithm with linear time complexity that maintains the accuracy of less scalable clustering approaches. Linear time scaling requires employing only linear time algorithms, such as radix sorting. To achieve this goal, the algorithm exploits the underlying relationships among sequences−i.e., the distance from A to C and A to B reveals something about the approximate distance between B and C. This allows the sequences to be sorted in a manner analogous to the ordering of leaves along a phylogenetic tree. Hence, after pre-sorting, nearby sequences are more likely to be closely related, and distant sequences can be ignored. This new approach, named Clusterize (i.e., a blend of cluster + linearize), is implemented within the DECIPHER package[17] for the R programming language[18] and available as part of the Bioconductor package repository[19].

## Results

### Overview of the Clusterize algorithm

General-purpose clustering algorithms are expected to handle homologous and non-homologous groups of input sequences with considerable variability in group sizes (i.e., imbalance). In phase 1, the Clusterize algorithm separates input sequences into partitions of detectable homology by counting rare k-mers shared between sequences. First, k-mers are randomly projected into a lower dimensional space using hashing. The number of k-mers in each hash bin is tabulated, and up to 50 (by default) k-mers corresponding to bins with the lowest frequencies are selected from each sequence. The resulting vector of rare k-mers is ordered in linear time using radix sorting, which yields groups of sequences that share a rare k-mer. This process is reflective of that used by Linclust, although it differs in the choice of hash function. Next, Clusterize tabulates the number of rare k-mers shared between a sequence and all others. Sequences sharing rare k-mers are recorded starting from the rarest k-mer and continuing until a user-specified limit (parameter *A*, by default 20,000) on the number of sequences is reached.

The number of rare k-mers shared per sequence tends to be near zero for unrelated sequences. To determine the background distribution, Clusterize fits a curve to the initial exponential decay in the number of rare k-mers shared between sequences (Fig. 1A). This permits related sequences to be identified because sequences are unlikely to share many rare k-mers by chance. Sequences sharing a sufficient number of rare k-mers are added to the same partition (Fig. 1B), and a depth-first search is commenced to recruit other sequences to the partition (Fig. 1C). Repeating this process results in disjoint sets of sequences sharing a statistically significant number of rare k-mers with at least one other sequence in the set (Fig. 1D). These partitions are treated independently in phase 2 of the Clusterize algorithm.

The intuition underlying phase 2 is to pre-sort sequences within partitions into an order that is similar to where they would be located at the tips of a phylogenetic tree. That is, more similar sequences would be closer to each other after ordering. A straightforward way to arrange sequences in this manner is to imagine an unrooted phylogenetic tree with two leaves (Fig. 1E). A third leaf could be added at any point along the edge between the two existing leaves (Fig. 1F). Knowing the distances ($d_i$) between the third sequence and the other two ($d_1$ and $d_2$) helps to decide whether the new leaf belongs closer to one leaf than the other. If two new leaves are both much closer to one of the leaves, it is possible they are also close to each other (Fig. 1G). Ordering sequences in this manner can be thought of as relatedness sorting because more related sequences are closer together after sorting.

Choosing two leaves randomly and calculating their difference in distance ($d_1 − d_2$) to each sequence results in a vector of relative distances (Fig. 1H). This vector ($\boldsymbol{D_1}$) is then projected onto the axis of maximum variance shared with another relative distance vector ($\boldsymbol{D_2}$) calculated from two other random leaves (Fig. 1I). Projecting in this

manner is equivalent to performing principle components analysis on $\boldsymbol{D_1}$ and $\boldsymbol{D_2}$ to retain only the first principal component ($\boldsymbol{D}$). Repeatedly projecting new vectors onto $\boldsymbol{D}$ results in more related sequences having more similar relative distances and, thus, moving closer together in the ordering of $\boldsymbol{D}$. Sorting the distance vector therefore provides an ordering similar to that of the leaves along a phylogenetic tree. After several iterations, large gaps may appear in the relative distance vector that can be used to split the partition into smaller groups (Fig. 1J). Relatedness sorting is continued within each group until the ordering of sequences stabilizes within a tolerance or a maximum number of iterations (parameter *B*, by default 2,000) is reached (Fig. 1K).

Given that no sequences are clustered in phases 1 or 2, the objective of phase 3 is to establish linkage to a cluster representative as rapidly as possible. Each sequence only needs to be compared with a fixed number of sequences around it in the sequence ordering, resulting in a linear time clustering algorithm (Fig. 1L). If the sequence ordering is sufficiently out of order then potential links can be missed. To remedy these cases, Clusterize compares each sequence with sequences sharing the most rare k-mers, which are identified using the same process as in phase 1. This rare k-mer approach is analogous to that of Linclust except that sequences can be compared to multiple sequences per rare k-mer group rather than only one sequence per group. Calculation of k-mer similarity only needs to be performed on a fixed number of representatives (parameter *C*, by default 2000) corresponding to the clusters previously assigned to sequences that are nearby in the relatedness ordering or share rare k-mers. Clusterize keeps track of which approach (i.e., relatedness sorting or rare k-mers) results in the most clusters and proportionally draws more cluster representatives for comparison from the more rewarding approach.

Each phase requires up to a user-specified number of comparisons that each take linear time. In the end, a limited subset (parameter *E*, by default 200) of sequences with the highest k-mer similarity are aligned to obtain their percent identities. Therefore, the complete algorithm has time complexity $O(A{*}N) + O(B{*}N) + O(C{*}N) + O(E{*}N)$ with respect to *N*. When *A*, *B*, *C*, and *E* are bounding constants, the asymptotic time complexity becomes $O(N)$. The amount of memory required scales similarly, $O(N)$, because only a fixed number of vectors of length *N* are required to store k-mers, distances, and orderings. As shown in Supplementary Fig. 1, parameters *A*, *B*, *C*, and *E* were optimized to minimize the number of clusters and time required using sequences from the RNAcentral database[20], which was chosen because of its large number of diverse sequences. Phase 3 often dominates clustering time and, accordingly, parameter *C* largely controls the trade-off between speed and accuracy. In contrast, parameters *B* and *E* typically have less effect because the algorithm converges prior to reaching the limit.

Clusterize uses a non-standard definition of k-mer similarity that offers some advantages. K-mer anchors are defined by the largest set of k-mers with consistent (i.e., collinear) ordering across a pair of sequences (Fig. 1M). For example, if region 50 to 100 of one sequence matches region 1 to 50 of a second sequence, then it is possible to exclude a 5-mer match between positions 1 to 5 and 55 to 59 because these k-mers conflict with the ordering of the larger matching region. K-mer similarity is then calculated as the number of positions in anchors divided by the estimated number of overlapping positions between the sequences. An added advantage of computing k-mer regions is that they can be used to constrain pairwise alignment (Fig. 1N), such that sequences can be aligned in sub-quadratic time ($< O(L^2)$)[21]. During pre-sorting (i.e., phase 2), Clusterize also aligns a small number of randomly sampled sequence pairs to determine the relationship between k-mer similarity and alignment similarity. This allows a boundary to be drawn, wherein sequences with too little k-mer similarity do not need to be aligned during phase 3 (Fig. 1O).

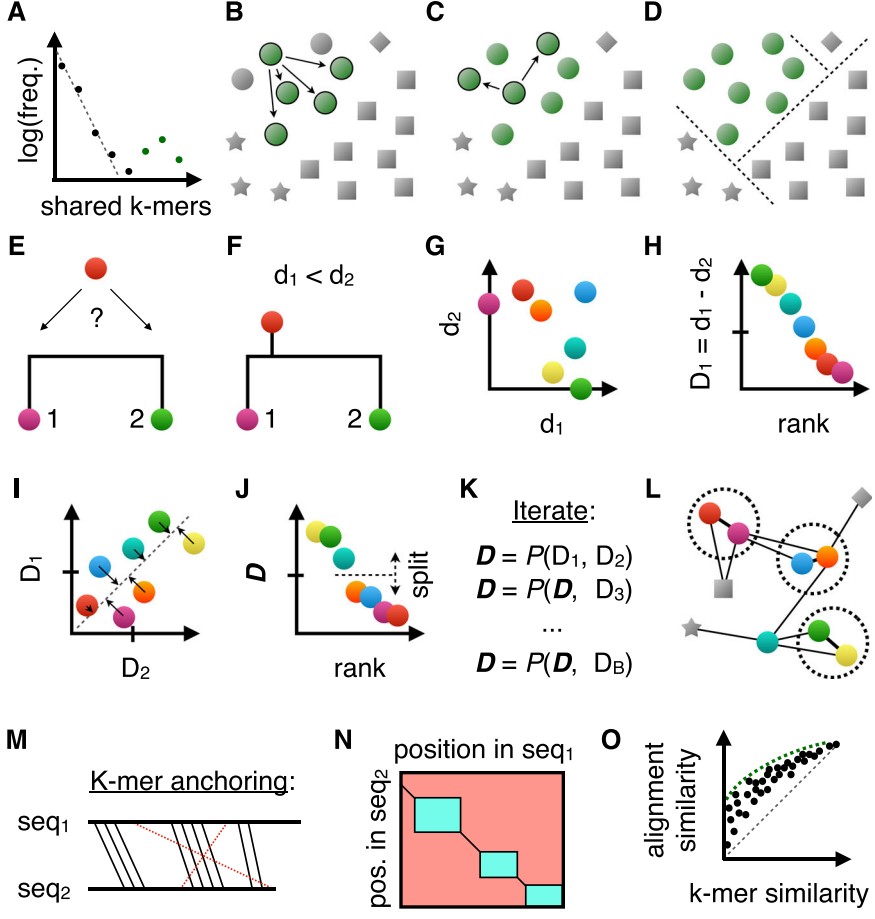

**Fig. 1 | Illustration of the Clusterize algorithm. A** In phase 1, the number of rare k-mers per sequence shared with a focal sequence are tabulated. The initial exponential fall-off in background frequency (dashed line) is used to identify related sequences (green points). **B** Statistically significant relationships are merged into the same partition. **C** This recruitment process is repeated for new sequences added to the partition. **D** Disconnected sets are used to define partitions of related sequences (shapes). **E** In phase 2, sequences within a partition are compared to two randomly selected input sequences from the partition using an alignment-free approximation of distance. **F** A sequence can be placed somewhere along the edge between the two sequences. **G** Sequences tend to be closer to the first or second random sequence, or sequences can be far from both (e.g., the blue dot). **H** The relative distance is used to sort the sequences by relatedness. **I** New relative distance vectors are projected onto the axis of maximum variance, *D*. **J** Partitions are split into separate groups when their rank order stabilizes locally. **K** Projection (*P*)

continues within each group until the rank order stabilizes or a bounding constant number of iterations is reached (parameter *B*). **L** In phase 3, the final rank order is used to determine which sequences to compare. Only a limited number of alignments (lines) are required per input sequence, including some sequences from outside the partition that share rare k-mers. Sequences within a user-specified similarity threshold are clustered (dashed circles). **M** To quickly approximate distance, K-mers are matched between sequences (thick horizontal lines) and chained into the longest set of anchor regions by rejecting inconsistently ordered k-mers (red). **N** Anchors (diagonal lines) are used to constrain the alignment space to the regions between anchors (green), thereby accelerating the calculation of alignment similarity. **O** During phase 2, a subset of sequence pairs are aligned to learn the maximum (green curve) alignment similarity expected for a given k-mer similarity. This allows sequences with low k-mer similarity to be ignored because they are unlikely to meet the clustering similarity threshold.

## Clusterize is comparable in accuracy to less scalable clustering programs

Clustering approaches are often compared by the number of clusters they generate at the same similarity threshold, although this comparison fails to take into account different definitions of similarity and linkage (e.g., average, complete, or single). Therefore, previous benchmarks have relied on the taxonomy[22] or ecology[23] of input sequences as proxies for biologically meaningful ground truth. The central assumption being that similar sequences will more often come from the same organism, sample, or habitat than dissimilar sequences. Under this assumption, it is possible to quantify accuracy using normalized mutual information (NMI), which measures the extent to which clusters and group labels agree[22]. NMI ranges from 0 to 1 and penalizes splitting or merging incorrectly[24]. A related measure, adjusted mutual information (AMI), additionally accounts for the possibility that observed and expected clusters could agree by random chance[25]. A more biologically meaningful clustering will have a higher AMI and

NMI with the biological ground truth at the same number of output clusters.

I sought to compare the accuracy of Clusterize with other clustering programs on large sets of protein sequences. To this end, I used 411 sets of proteins with more than 20,000 (i.e., 20,030 to 108,570) unique sequences matching families in TIGRFAM[26], which is a set of hidden Markov models for common prokaryotic genes. Each of these sequence sets was clustered at multiple thresholds between 40 and 100% similarity. At high similarity thresholds, Clusterize identified more clusters using rare k-mers, while at low similarity thresholds more clusters were identified with relatedness sorting (Supplementary Fig. 2). To measure accuracy, I calculated AMI and NMI using NCBI's class-, family-, and genus-level taxonomic assignment for each sequence. Clusterize returned clusters with AMIs and NMIs similar to MMseqs2 and UCLUST across the range of similarity thresholds (Fig. 2). There was a gap in peak AMI and NMI between the top three programs and Linclust or CD-HIT, and this gap was most pronounced

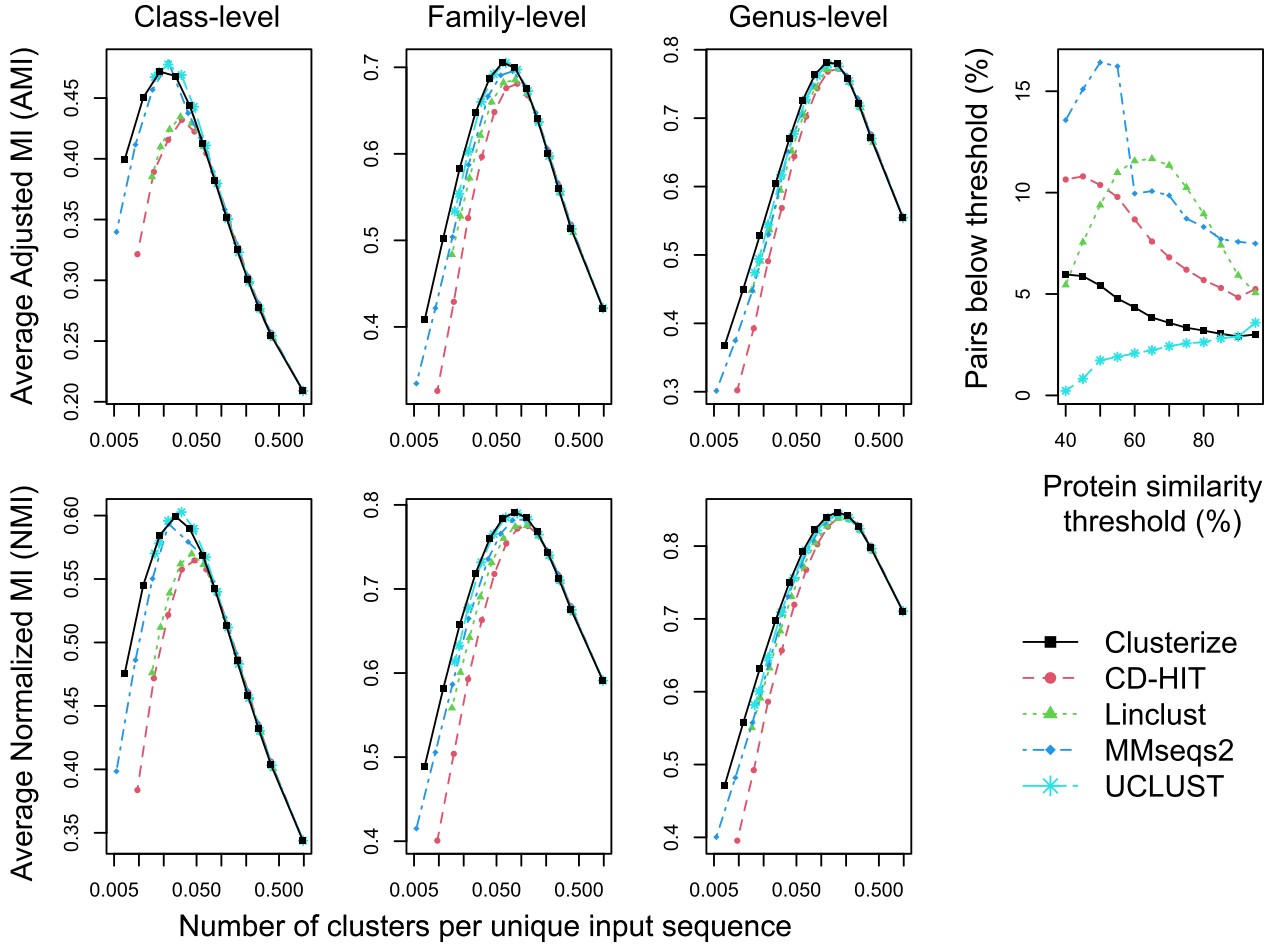

**Fig. 2 | Clusterize achieves high accuracy on 411 large TIGRFAM protein families.** AMI and NMI with class-level, family-level, and genus-level taxonomy were used to estimate accuracy when clustering large (>20,000 sequences) protein families. Each point represents average accuracy at a similarity threshold ranging from 40% to 100% in 5% increments (left to right, respectively). Note the log-scaled x-axis. Better accuracy for a rank-level appears as a higher peak. Differences among programs become more pronounced at the class-level. Notably, CD-HIT and Linclust returned lower accuracy clusters than the other programs at low similarity thresholds. The average fraction of clustered sequences falling below the similarity threshold is shown for up to 10,000 randomly selected sequence pairs per TIGRFAM protein family. CD-HIT, Linclust, and MMseqs2 clustered the most sequence pairs that failed to meet the specified similarity threshold.

at the class-level (Fig. 2). Given that Linclust is part of the MMseqs2-software suite, and differs from MMseqs2 in the use of heuristics to scale linearly, this result indicates Linclust's heuristics decrease accuracy.

Inexact algorithms typically establish linkage to a single representative sequence per cluster but may employ different heuristics to accelerate this process. Therefore, it is expected that randomly selected pairs of sequences from each cluster are usually, but not always, within the specified similarity threshold. To test this expectation, I calculated the fraction of up to 10,000 randomly selected pairs of clustered sequences that obeyed each similarity threshold. Averaged across the 411 TIGRFAM protein families, about 3 to 6% of sequences belonging to the same cluster exceeded the similarity threshold given to Clusterize (Fig. 2). A similar behavior was observed for UCLUST, except at low similarity thresholds (<50%) where almost no clustered sequences exceeded the similarity threshold. In contrast, CD-HIT, Linclust, and MMseqs2 frequently did not obey the similarity threshold, with about 10 – 18% of clustered sequences exceeding the threshold at intermediate or low similarity thresholds. These results imply that CD-HIT, Linclust, and MMseqs2 can form larger clusters than Clusterize and UCLUST at the same similarity threshold, likely reflecting a consequence of the heuristics employed by each program. For example, MMseqs2 converts sequence identity into an alignment score per residue for clustering, implying the percent identity of

clustered sequences is expected to deviate from the user-specified similarity threshold.

I repeated this analysis on 3,001 TIGRFAM protein families with fewer sequences (i.e., 2 to 20,352), for which it was feasible to perform exact hierarchical clustering. For each sequence set, a multiple sequence alignment was constructed with DECIPHER[27] in order to obtain a distance matrix for average-linkage and complete-linkage clustering. The inexact clustering of smaller TIGRFAM protein families closely reflected the results previously observed for larger families (Supplementary Fig. 3). UCLUST and Clusterize were the top two programs, followed by MMseqs2, Linclust, and CD-HIT. However, both average-linkage and complete-linkage clustering outperformed all inexact methods in AMI and NMI across all rank levels (Supplementary Fig. 3). Exact clustering also clustered the fewest sequences beyond the similarity threshold, further confirming the superiority of exact over inexact clustering. Collectively, clustering TIGRFAM protein families revealed the merits of different clustering approaches, but none of the TIGRFAM protein families contained enough sequences to estimate programs' asymptotic behaviors.

## Clusterize accurately clusters millions of SARS-CoV-2 genomes in linear time

A common use of clustering programs is to reduce the size of very large sets of nucleotide sequences, such as protein-coding gene or

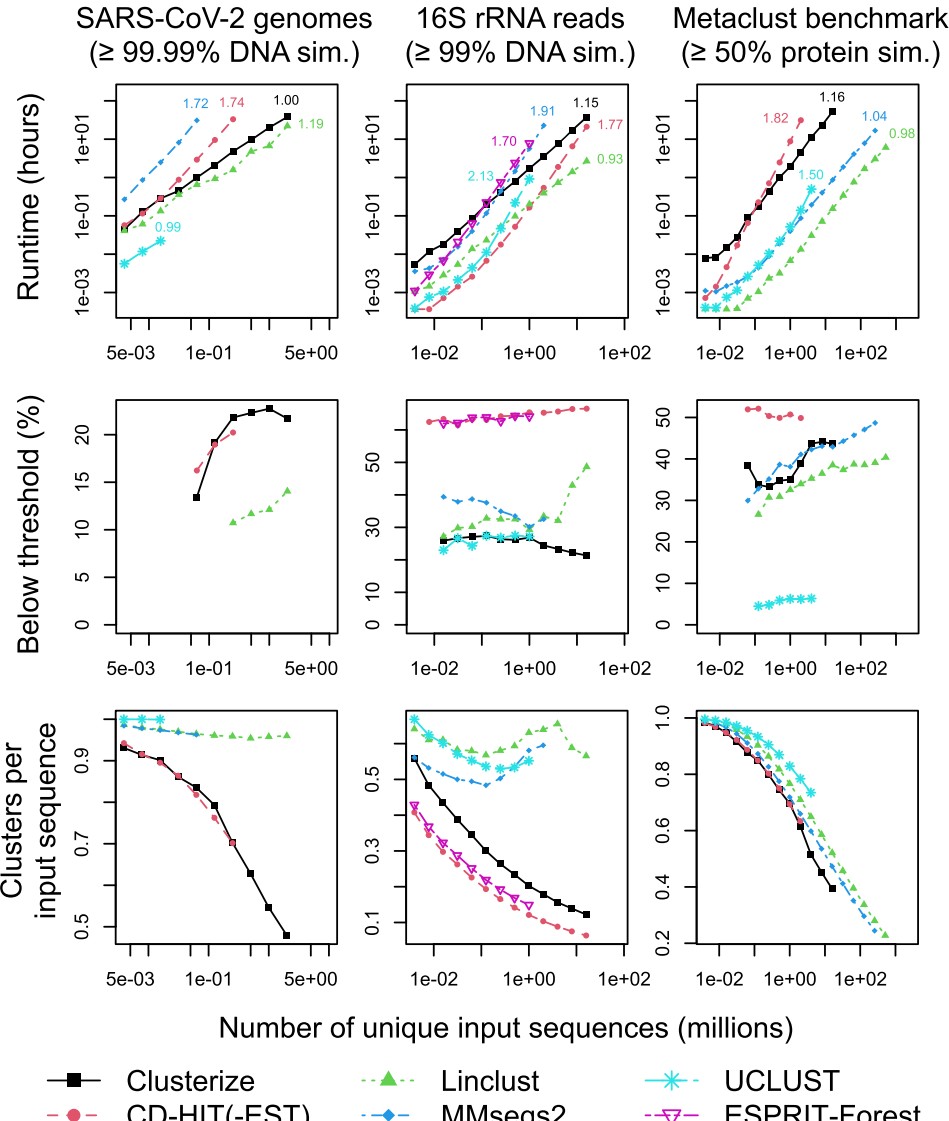

**Fig. 3 | Comparing programs on highly varied clustering tasks.** Complete or nearly full-length (>29,000 nucleotides) SARS-CoV-2 genomes were clustered at 99.99% nucleotide similarity. Only Clusterize and Linclust clustered two million genomes within the time limit. Inset numbers represent the order ($x$) of the scaling polynomial (i.e., $O(N^x)$) estimated from the largest four inputs on each curve to approximate an algorithm's asymptotic empirical time complexity. UCLUST formed almost exclusively singleton clusters with SARS-CoV-2 genomes, effectively failing to cluster the sequences. A random sample of up to 10,000 pairs of clustered sequences was drawn from each program's clusters to estimate the percentage of clustered sequences below the specified similarity threshold. Programs with fewer than 5000 clustered sequences at a given input size are not shown. An exact algorithm, ESPRIT-Forest, was included when clustering up to 16 million 16S rRNA sequences collected from healthy human subjects in the Human Microbiome Project. Linclust was the fastest program but also returned the smallest clusters, suggesting it failed to adequately cluster the sequences. CD-HIT-EST and ESPRIT-Forest formed the fewest 16S clusters per sequence but also least obeyed the specified similarity threshold. Linclust and MMseqs2 were the fastest programs on the Metaclust benchmark composed of metagenome protein fragments. However, Linclust and MMseqs2 produced more clusters than Clusterize despite clustering a similar fraction of sequences below the 50% similarity threshold.

non-coding RNA sequences. To investigate the use of Clusterize for this purpose, I downloaded all 13 million SARS-CoV-2 genomes available from the GISAID initiative[28], and selected the subset of 2 million unique genomes with no ambiguities (e.g., "N") and lengths between 29,000 and 30,000 nucleotides. Gradually increasing numbers of sequences were clustered at 99.99% similarity until a program failed or was projected to require more than 48 hours to cluster the sequences. A very high similarity threshold was required because SARS-CoV-2 genomes typically differ at only a few positions. Both Clusterize and Linclust displayed approximately linear asymptotic scalability with the number of input sequences, while CD-HIT-EST and MMseqs2 scaled superlinearly (Fig. 3).

As more sequences are clustered, it is reasonable to assume the number of clusters per sequence will decrease because there is additional redundancy among the sequences. A decreasing trend was observed for all programs except UCLUST, which almost exclusively created singleton (i.e., unclustered) clusters (Fig. 3). The fact that UCLUST effectively did not cluster the sequences was unexpected and suggests an issue with UCLUST's parameters when clustering long sequences. Surprisingly, the other programs generated very different numbers of clusters from the same input sequences (Fig. 3). Linclust and MMseqs2 produced a similar number of clusters per input sequence regardless of the number of input sequences, suggesting they failed to adequately cluster the sequences. CD-HIT-EST and Clusterize created the fewest clusters and cluster sizes steadily

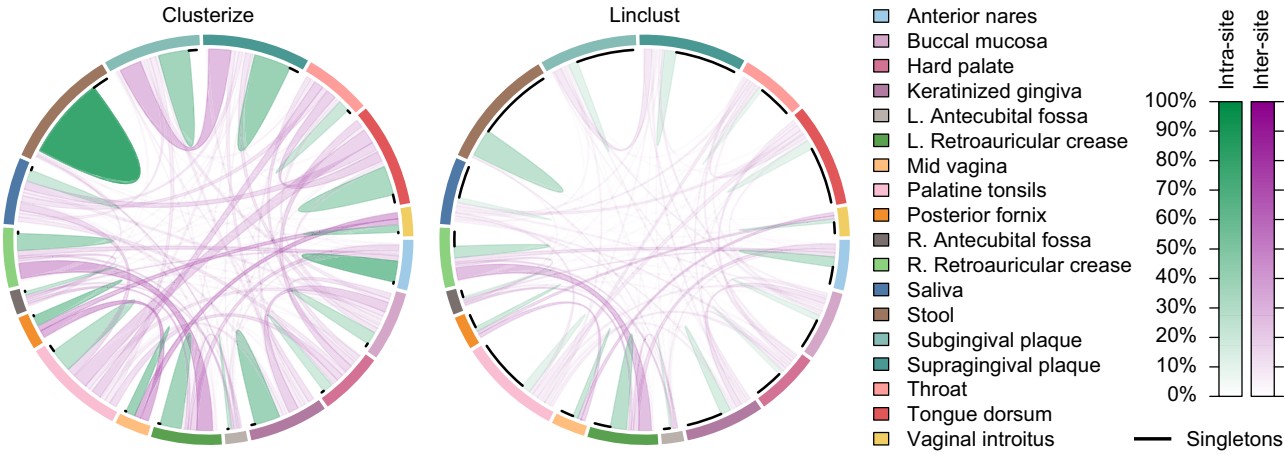

**Fig. 4 | Biological interpretability of clustering 16 million microbiome reads.** Chord diagrams show the clustering of 16S rRNA sequences obtained from different sites on the human body (sectors) as part of the Human Microbiome Project. Each sequence is assigned connectivity (chords) in proportion to the number of sequences from each body site present in its cluster, and connections are shaded in proportion to the fraction of reads from each site. Clustered sequences belonging to the same body site are shown in green, connections between different body sites are shown in purple, and singletons are shown as black curves. Clusterize produced far fewer singleton clusters than Linclust when clustering the same set of 16 million sequences at 99% similarity. The correlation in clustered sequences across body sites is clearer in the Clusterize output. Body sites that are expected to have similar microbial communities, such as different parts of the mouth, are more interconnected by Clusterize than Linclust.

increased with more input sequences. With 2 million input sequences, only Clusterize and Linclust completed in the allotted time, producing 957,945 and 1,919,893 clusters, respectively. Taken together, these results demonstrate the extended reach of Clusterize's accuracy because of its linear time complexity.

Several steps of the Clusterize algorithm are parallelized with OpenMP, including integer encoding k-mers, matching k-mers across sequences, and aligning the regions between k-mer anchors. To test the efficiency of parallelization, I clustered a fixed number of sequences with increasing numbers of processors. At maximum, Clusterize achieved about a 10-fold speedup using 28 processors, which was greater than the speedup achieved by CD-HIT-EST or Linclust but less than MMseqs2 (Supplementary Fig. 4). UCLUST offered almost no speedup using multiple processors, which was consistent with the UCLUST documentation. Both Linclust and MMseqs2 returned clusters with different numbering when using multiple processors, although the underlying clusters were the same. UCLUST produced different clusters when using multiple processors (Supplementary Fig. 4), but the results were repeatable with the same number of processors. Hence, reproducibility with UCLUST necessitates specification of the number of threads. Clusterize, although a stochastic algorithm, always returned the same results when the random number seed was set, regardless of the number of processors (Supplementary Fig. 4).

### Clustering millions of homologous microbiome sequences across samples

The 16S ribosomal RNA (rRNA) gene is commonly used as a phylogenetic marker in microbiome studies and is sometimes analyzed in a reference-independent manner by clustering into OTUs. Thresholds for clustering typically range from 97% to 100% sequence similarity depending on the length of sequences being clustered and desired clustering granularity[29,30]. Clustering is particularly useful for comparing the compositions of microbiome samples collected from different habitats. To compare clustering programs for this purpose, I downloaded the set of 24 million unique quality-trimmed reads originating from different body sites on healthy human subjects in the Human Microbiome Project that were amplified with V3-V4 16S primers[31]. Sequences were clustered at 99% sequence identity to create OTUs, which were then compared across body sites where the samples

were collected. I included an additional clustering program, ESPRIT-Forest, that was designed specifically for exact hierarchical (average-linkage) clustering of 16S OTUs with sub-quadratic (i.e., $<O(N^2)$) time complexity[15].

Clusterize and Linclust displayed approximately linear time asymptotic scaling, while the other programs had near-quadratic scaling (Fig. 3). ESPRIT-Forest scaled subquadratically, but errored unexpectedly before reaching the time limit. Linclust, MMseqs2, and UCLUST all generated far more clusters per input sequence than Clusterize, CD-HIT-EST, or ESPRIT-Forest, implying they failed to cluster many sequences that could have been clustered. CD-HIT-EST and ESPRIT-Forest created the fewest clusters but also had the broadest clusters, with over 60% of clustered sequences failing to meet the specified similarity threshold (Fig. 3). In contrast, Clusterize and UCLUST produced clusters with fewer than 30% of clustered sequences beyond the similarity threshold.

Although MMseqs2 created about the same number of clusters as Clusterize for the lowest input size (3,906 sequences), the two programs diverged markedly with larger input sizes (Fig. 3). Clusterize produced 1,954,730 clusters at the largest input size, in sharp contrast to Linclust's 9,050,721 clusters. Also, with 16 million input sequences, Clusterize assigned 9.2% of clustered sequences exclusively with relatedness sorting and 27.9% exclusively with rare 13-mers. As shown in Fig. 4, the correlation among sequences obtained from different body sites was much clearer with Clusterize than Linclust due to the relative rarity of singletons. This result depicts the practical implications of higher clustering accuracy for deriving biologically meaningful results.

### Clusterize was slower but more accurate than Linclust on Metaclust

The authors of Linclust introduced the Metaclust benchmark, which contains more than a billion protein fragments derived from metagenomic and metatranscriptomic datasets. I sought to compare each program's performance for the authors' original goal of clustering protein sequences to increase search accuracy and speed through very large sequence sets. To this end, I downloaded the full set of Metaclust sequences, removed exact duplicates, and randomized their order before clustering at a low (50%) similarity threshold. Both Clusterize and CD-HIT created the fewest clusters per input sequence, although

CD-HIT clustered more sequences beyond the 50% similarity threshold (Fig. 3). Clusterize, Linclust, and MMseqs2 had comparable rates of clustering sequences outside of the similarity threshold, although Clusterize generated fewer clusters. With 16 million input sequences, Clusterize generated 6,285,025 clusters relative to MMseqs2's 7,567,092 and Linclust's 8,339,747 clusters, corresponding to 1,282,067 (20%) and 2,054,722 (33%) more clusters than Clusterize, respectively. However, MMseqs2 and Linclust were the fastest programs, which allowed them to cluster far more sequences than the other programs within the allotted time. These results further highlight the speed versus accuracy trade-off when choosing a program for a specific clustering application.

### Reducing redundancy among tens-of-millions of diverse sequences

Large sequence databases are frequently clustered to reduce the number of sequences to a more manageable size for various applications. An example of this is UniRef[32], which provides clusters of Uni-Prot sequences at three similarity thresholds (50, 90, and 100%). Comparing programs for this purpose requires an extremely large and diverse collection of sequences that are accurately annotated with biologically meaningful groups. To this end, I downloaded all unique protein (amino acid) and protein-coding (nucleotide) sequences available from 44,831 randomly selected bacterial genomes from the NCBI Reference Sequence Database (RefSeq). I reasoned that protein functional classifications could be used to measure clustering accuracy in much the same way as taxonomy. Protein sequences were labeled by their predicted function and hypothetical proteins were discarded. These protein functional classifications were then transferred to the protein-coding sequences to generate matched sets of 151,835,459 labeled protein and nucleotide sequences for benchmarking.

As shown in Fig. 5, programs were compared at protein similarity thresholds of 50% and 100%, as well as nucleotide similarity thresholds of 80 and 100%. As expected, Clusterize and Linclust displayed approximately linear asymptotic scaling in runtime, although Linclust was 7- to 18-fold faster at clustering 32 million nucleotide sequences and 19- to 71-fold faster at clustering 64 million protein sequences. MMseqs2 also showed near linear runtime and was almost as fast as Linclust for protein sequences. CD-HIT and UCLUST had super-linear asymptotic scaling in runtime, which made them the slowest programs in many cases. Consistent with the previous benchmarks, Linclust and MMseqs2 clustered the most sequences below the specified similarity threshold (Fig. 5). Given the use of only two similarity thresholds per sequence set, I chose to measure accuracy with a previously published approach based on the average consistency of labels among clustered sequences[33]. Clusterize consistently returned clusters with higher average label consistency than MMseqs2 or Linclust (Fig. 5).

Memory consumption imposes limitations beyond time constraints when clustering many sequences. To track peak memory usage, I recorded the maximum amount of main memory used by each program. Both Clusterize and Linclust consumed memory proportional to the number of input sequences, with Clusterize requiring about 1.4-fold more memory for clustering 32 million nucleotide sequences and 1.6-fold more memory for clustering 64 million protein sequences (Fig. 5). MMseqs2 reserved a large amount of memory for small numbers of sequences but scaled similarly to Linclust for large input sizes. The freely available (32 bit) version of UCLUST was limited to 4 gigabytes of memory, which prevented UCLUST from clustering some input sizes that the 64 bit version would likely have finished before the time limit.

## Discussion

The continually increasing availability of sequences demands scalable clustering algorithms. To my knowledge, Clusterize is the first use of relatedness sorting to linearize clustering. Furthermore, the deployment of an ordered k-mer-based measure of similarity allows Clusterize to employ anchored alignment that makes aligning long sequences more scalable. Collectively, these features enable the Clusterize algorithm to cluster millions of homologous or non-homologous sequences with high accuracy. Clusterize was slower than many programs for small sets of sequences but sometimes faster for large numbers of sequences. Unexpectedly, the degree of inexactness varied markedly across programs and applications. While Clusterize consistently conformed to expectations, other programs sometimes failed to adequately cluster sequences or obey the user-specified similarity threshold. This result suggests some heuristics are substantial compromises for increased speed, and users may unwittingly obtain sub-optimal results with some programs.

A user of clustering programs might be tempted to associate speed with clustering quality, although the results of this study dispel any such relationship. Linclust was the fastest program but often produced smaller clusters with comparatively low accuracy. MMseqs2 and UCLUST were sometimes more accurate than Linclust, but their super-linear time complexity in some cases made clustering millions of sequences less practical. Clusterize showed high accuracy and linear scalability but was consistently slower than Linclust. For huge numbers of non-homologous sequences (>100 million), MMseqs2 or Linclust remain the most practical options for clustering in a reasonable amount of time, although potentially at the expense of accuracy. In a sense, there is no panacea, and users will need to weigh their priorities when deciding which program to choose. Clusterize excels at high accuracy clustering under its default settings and allows users to balance the runtime versus accuracy trade-off by adjusting parameters (Supplementary Fig. 1). In contrast to Clusterize, Linclust's accuracy is limited by the number of k-mers per sequence, which was set to the high value of 80 in this study to improve Linclust's accuracy over its default of 20 k-mers per sequence.

The scalability of clustering programs with non-linear time complexity was dependent on the number and diversity of sequences, as well as the similarity threshold. Strategies with $O(M*N)$ time complexity are most efficient when the similarity threshold decreases to the point where the entire diversity of sequences fits within few clusters (i.e., $M$ approaches 1). In the opposite regime, when there are many clusters, strategies with $O(N)$ time complexity become important as $N$ becomes large. Clusterize relies on two $O(N)$ strategies, relatedness sorting and rare k-mers, and outputs which strategy resulted in the most clustered sequences at completion. With default settings, rare k-mers are typically the source of more clusters when the similarity threshold is high, whereas relatedness sorting is typically the source of more clusters when the similarity threshold is low (Supplementary Fig. 2). However, there are many input sequence sets and similarity thresholds that cause Clusterize to source clusters from a combination of both strategies.

Clusterize's use of rare k-mers differs considerably from that of Linclust. By default, Clusterize compares each sequence against up to 2000 (i.e., parameter $C$) sequences sharing the greatest number of rare k-mers. In contrast, Linclust compares each sequence to the longest sequence sharing a rare k-mer. Nevertheless, both approaches perform worse as the number of sequences sharing rare k-mers becomes very large, such as when clustering many homologous sequences. This happens to be the case where relatedness sorting performs especially well, so Clusterize's two linear time approaches are complementary and allow it to tackle many clustering scenarios. Clusterize is designed to automatically detect which approach is working best and draw a greater proportion of the 2000 (i.e., parameter $C$) sequences from the better approach in phase 3. This enables Clusterize to rely on relatedness sorting when rare k-mers fail and vice versa.

It may initially appear surprising that relatedness sorting is useful. By design, relatedness sorting projects similarity among sequences into a single dimension, and distant sequences could be

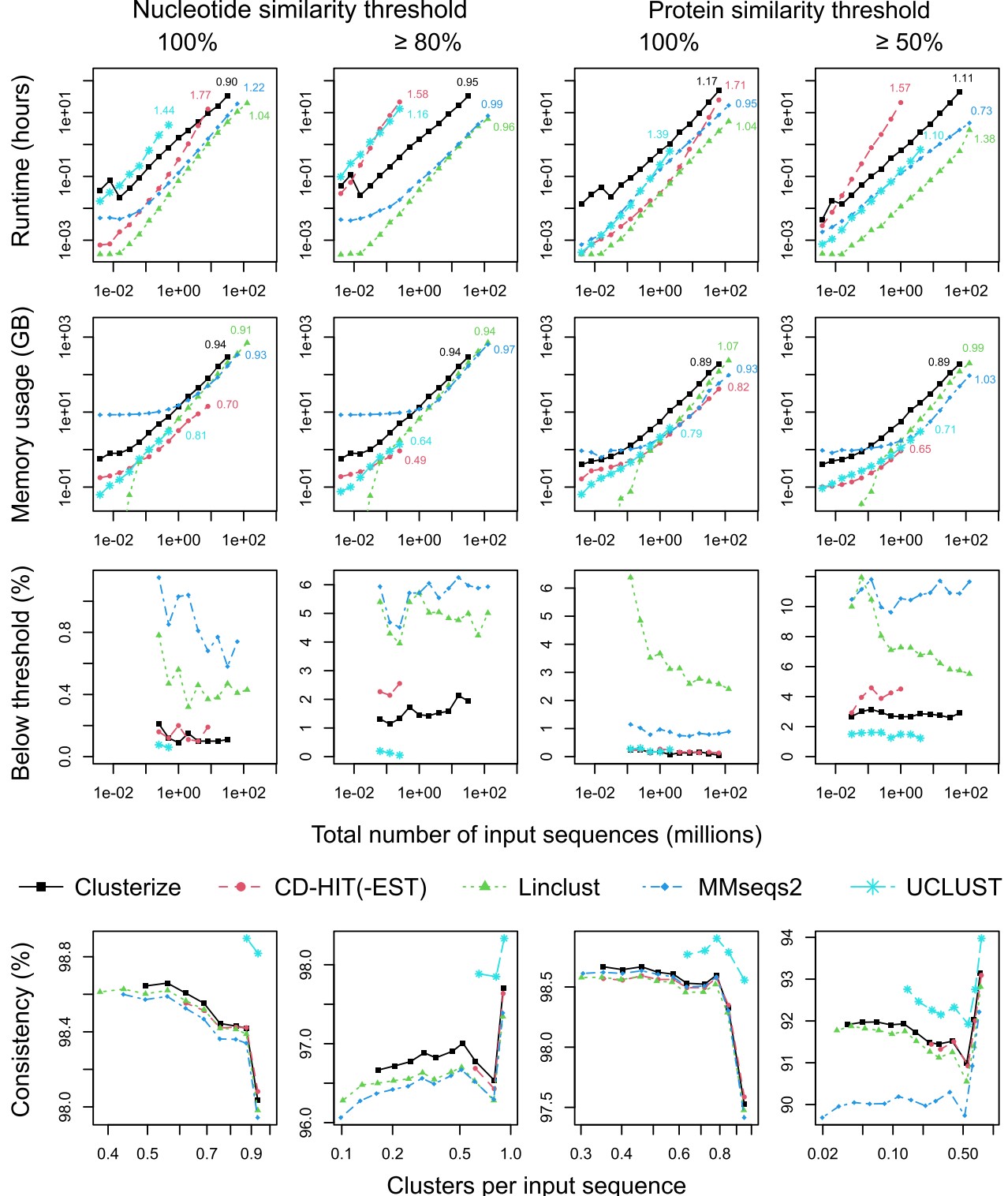

**Fig. 5 | Clustering up to 128 million protein-coding and protein sequences.**
Functionally annotated protein-coding (nucleotide) and protein (amino acid) sequences originating from 44,831 bacterial genomes were clustered at different similarity thresholds. Inset numbers give the order (*x*) of the scaling polynomial (i.e., $O(N^x)$) estimated from the largest four inputs on each curve. Linclust and MMseqs2 were the fastest programs for the largest input sequence sets. Clusterize required about 1.5-fold more memory than MMseqs2 and Linclust when clustering 32 million nucleotide or 64 million protein sequences. The free (32 bit) version of UCLUST was limited to 4 GB of memory and therefore failed to complete some input sets that it may have otherwise completed before the time limit. Linclust and MMseqs2 clustered the most pairs of sequences under the specified similarity threshold. Clusterize produced clusters with higher label consistency than CD-HIT(-EST), Linclust, or MMseqs2. UCLUST returned clusters with the highest label consistency, but also generated smaller clusters than the other programs.

projected near each other on a one-dimensional line. The effectiveness of this approach becomes clearer when considering a phylogenetic tree. All homologous sequences, no matter how distant, can be arranged along the tips of a tree and projected onto a single line. Sometimes distant clades will be placed adjacent to each other along the projection, but each sequence will generally be adjacent to its most similar neighbors. The Clusterize algorithm works by iteratively constructing this projection through relatedness sorting without needing to construct the entire tree. Phase 1 globally partitions the sequences by homology, phase 2 conducts relatedness sorting within each group, and phase 3 performs a local similarity search for the nearest neighbor. This clustering strategy could be applied to any set of related objects, although the focus of this study was clustering biological sequences.

Gauging the accuracy of different clustering outputs at scale is challenging. Simulations have been used to generate sequence sets with a known ground truth[34,35], but simulations necessitate many assumptions. Here, I chose to stay consistent with recent clustering benchmarks by using AMI and NMI with a taxonomic group (Fig. 2 and Supplementary Fig. 3) as a measure of accuracy[22,23]. The strong signal of vertical inheritance in gene trees[36] makes taxonomy a reasonable proxy for capturing biologically meaningful similarities within homologous groups, which is often the objective of sequence clustering. Similarly, functional annotations can be used to judge clustering accuracy for non-homologous sequences, especially at low similarity thresholds. An alternative approach is to compare with the results of exact hierarchical clustering, which is known to result in biologically meaningful clusters but is less scalable[22,23,25,37]. The results of exact hierarchical clustering on the set of small TIGRFAM protein families (Supplementary Fig. 3) show there is considerable room for improvement beyond inexact methods that establish linkage to only a single sequence per cluster.

Clustering is sometimes used to define homologous groups of sequences, such as protein families. In my opinion, the task of defining homologous groups is poorly suited to inexact clustering at fixed identity thresholds for several reasons. First, homologous proteins sometimes share very little sequence identity, which can result in a failure to detect homology. Even using hidden Markov models for sequence recruitment can result in homology search failures[38]. Second, percent identity is a weak proxy for homology at low levels of similarity, making defining a clustering threshold problematic[39,40]. Better scoring measures have been developed for detecting homology at low sequence identities[41]. Third, natural groups of sequences should be expected to deviate from fixed thresholds, and flexible thresholds are possibly more amenable to identifying natural group boundaries[42]. A recent attempt was made to circumvent some of these limitations by clustering protein structures instead of sequences[43]. Nevertheless, similar protein folds could result from convergent evolution rather than homology, and sequences remain essential for determining shared ancestry[44].

Notwithstanding these generic limitations of clustering, there are several applications where inexact clustering is particularly useful, some of which were used here to compare programs. In these situations, and others, clustering is an essential tool in the toolbox of bioinformatics algorithms. Exponential growth in the number of publicly available sequences, with a doubling time of approximately 3 years, will steadily increase the necessity of linear time clustering algorithms. The Clusterize algorithm can cluster tens-of-millions of sequences in reasonable amounts of time and does so more accurately than Linclust, although Linclust was much faster for huge sequence sets. Clusterize also returned results that consistently met expectations across a wide range of similarity thresholds, sequence lengths, sequence diversities, and input sizes. Therefore, Clusterize provides a dependable clustering algorithm that is intended to work well in many different user scenarios.

## Methods

### Measures of similarity between sequences

Clusterize permits several parameterizations of similarity. Most clustering programs define similarity as the number of matching positions divided by the length of the shortest sequence in each pair. By default, Clusterize defines similarity as the number of matching positions divided by the length of the overlapping region between pairs of sequences, because this definition works well for partial-length sequences. However, this definition is susceptible to deriving estimates of similarity from very small overlapping regions, which is why a coverage of at least 50% of positions is required by default. Clusterize allows the user to control whether gaps (i.e., insertions or deletions) are ignored, counted as mismatches or, by default, treated as a single mismatch per run of consecutive gaps. The latter definition represents an event-based model of distance, wherein a gapped region is considered a single event, analogous to a single substitution event. In practice, most definitions of distance yield comparable accuracies (i.e., AMI and NMI) at the same number of clusters, although the same number of clusters will be obtained at different similarity thresholds.

Since pairwise alignment is time-consuming, estimates of similarity are initially based on k-mer matching between pairs of sequences in phases 2 and 3. First, each sequence is converted into 32 bit integer vectors representing every overlapping k-mer. Second, these integers are ordered using radix sorting, and the k-mers' original positions are also stored as integers. These two integer vectors per sequence typically consume the majority of memory used by Clusterize. Third, the sorted integer vectors are matched between two sequences, and a vector is constructed mapping matching positions in one sequence to the other. Fourth, contiguous blocks of matching positions are recorded. Dynamic programming is used to determine the largest collinear set of blocks shared between the two sequences, defined as the maximum length set of matching k-mer regions (Fig. 1M). Fifth, these anchor regions are used to compute k-mer similarity or accelerate pairwise alignment (Fig. 1N). Sequences between anchor regions are aligned using the PFASUM50 substitution matrix[45] for amino acids or a scoring scheme of matches = 3, transitions = 0, and transversions = −3 for nucleotides. Standard affine gap penalties are added with gap opening = −10, gap extension = −2, and terminal gaps having zero cost during alignment.

A corresponding definition of k-mer similarity is possible for each user-defined parameterization of similarity but based on matching regions rather than positions. That is, anchor regions are considered matches, and unanchored regions are considered mismatches. The differences in distance between neighboring anchors in both sequences is used to determine the minimum number of gaps required to align the two sequences. Consistent with the default definition of similarity, k-mer similarity is defined as the number of matches divided by the number of overlapping positions plus the number of gapped regions. In phase 2, at least 1000 randomly selected sequence pairs are aligned to determine the relationship between k-mer similarity and percent identity. A logistic regression model is fit to predict the probability of percent identity meeting the user-specified similarity threshold given the k-mer similarity. In phase 3, up to 200 (i.e., parameter $E$) cluster representatives with k-mer similarity sufficient to have at least a 1% predicted probability (by default) of meeting the similarity threshold are aligned with each sequence. Cluster representatives with k-mer similarity already satisfying the similarity threshold do not need to be aligned because k-mer similarity will always be less than or equal to percent identity after alignment.

### Selection of optimal k-mers and k-mer length

Multiple sets of integer encoded k-mers are used throughout the Clusterize algorithm. In phase 1, amino acid k-mers are formed in a reduced alphabet. A similar strategy to that used for Linclust was employed to construct a 10-letter alphabet, except using the

PFASUM50[45] substitution matrix rather than BLOSUM62, which resulted in the merger of (A, S, T), (R, K, Q), (N, D, E), (I, V, L, M), and (F, Y). While a reduced alphabet is used to identify related sequences and partitions with rare k-mers, the standard (20 letter) amino acid alphabet is used for computing k-mer similarity because matches need to be exact.

Two different values of k-mer length ($k$) are computed: one for use with k-mer matching between sequences and the other for use with rare k-mers. The goal of selecting $k$ for k-mer matching is to ensure that k-mer matches between two sequences happen infrequently by chance, while using only a single value of $k$ across all sequences. Hence, $k$ is calculated such that one in every 100 k-mers is expected to be found by chance in sequences at the 99th percentile of input sequence lengths ($w_{99}$) using an alphabet with a given number of letters ($x$):

$$k = \lceil \log_x(100 * w_{99}) \rceil$$

The reasoning behind this formula for $k$ was previously described[46]. In contrast, $k$ for rare k-mers is set such that at most 10% of the selected 20,000 (i.e., parameter $A$) sequences are expected to share a rare k-mer due to chance. Each of $N$ sequences contributes 50 (by default) rare k-mers, and the objective is to find $k$ such that each rare k-mer will be found by chance in less than 40 (i.e., 20,000/50 * 10%) sequences on average. Thus, $k$ for rare k-mers is calculated as:

$$k = \lceil \log_x(50 * N/40) \rceil$$

The choice of which rare k-mers to select from each sequence is critically important[16]. Ideally, rare k-mers should be repeatedly selected such that (*i*) the same rare k-mers are chosen from similar sequences, (*ii*) rare k-mers are not positionally co-located on the sequence, and (*iii*) rare k-mers are relatively rare so that the same rare k-mers are more likely to be found in more similar sequences. Hashing functions meet these criteria as they allow for rare k-mers to be repeatedly identified across sequences[16]. I found that many different hashing functions were sufficient, and decided to use a standard 32-bit xorshift random number generator for this purpose[47]. The input integer encoded k-mer is used as a seed to produce a 32-bit random number, and an integer remainder is output after integer division by the size ($s$) of the output hash space (i.e., the modulo operation). Optimal performance was found where the size of the hash space equaled the square root of the number of possible distinct k-mers (i.e., $s = x^{k/2}$) or the mean sequence length (i.e., mean of $w$), whichever was greater. This choice of $s$ is intended to prevent many k-mers in the same sequence from being assigned to the same hash bin for very long input sequences.

Thus, each k-mer is randomly projected into the hashing space, and the frequency of each hash bin is tabulated. Up to 50 k-mers corresponding to the least frequent hash bins are selected from each sequence. This vector is ordered with radix sorting, yielding groups of sequences sharing a rare k-mer in much the same manner as described for Linclust[16]. Three vectors of length 50*$N$ are used to store the index of each sequence in the k-mer group, the starting position of the k-mer group, and the ending position of the k-mer group. In this manner, it is possible to efficiently find all sequences sharing a rare k-mer with a particular sequence by looking up the sequences belonging to each of its rare k-mer groups.

## Phase 1: separating sequences into partitions with detectable similarity

Prior to relatedness sorting it is necessary to first separate the input into partitions containing sequences sharing detectable homology. To this end, up to 20,000 (i.e., parameter $A$) sequences sharing a rare k-mer are selected for a given sequence. This is accomplished by sorting the sequence's 50 rare k-mers by their respective group sizes,

and recording all sequences in each rare k-mer group starting from the smallest group until up to 20,000 sequences are recorded. Often there are fewer than 20,000 sequences from all 50 rare k-mer groups, but this is not always the case and the maximum number of shared rare k-mers might be less than 50. The resulting 20,000 or fewer sequences are then tabulated for the number of times they are observed sharing a rare k-mer with the given sequence. These counts are tabulated to produce a vector of up to length 50 containing the number of sequences sharing that many rare k-mers.

Unrelated sequences typically share very few rare k-mers, and there is an initial exponential fall-off in the number of sequences sharing more-and-more rare k-mers by chance (Fig. 1A). A line is fitted to the initial fall-off in log-space, using points decreasing monotonically from the first local maximum to minimum. This line approximates the number of sequences in the input expected to share some number of rare k-mers by chance. The point at which the line goes below 1 is converted into a probability by subtracting from 1. For example, if 0.05 sequences are estimated to share 11 rare k-mers, then the probability of homology assigned to these sequences is 0.95. Sequences are sorted from the highest to lowest probability of homology (i.e., most to fewest shared rare k-mers), and the cumulative product is applied to the probabilities as a form of multiple testing correction. Any sequences with a remaining probability of at least 0.9 are deemed homologous to the given sequence. In practice, all sequences sharing many rare k-mers (>20 of 50) are considered homologous and those sharing very few (<10) are not.

It may seem counterintuitive that a sequence sharing 9 rare k-mers would not be considered homologous, given that rare k-mers are selected partly for their rarity. However, since the output hash space ($s$) is relatively small and up to 50 k-mers are selected, it is expected that some rare k-mers are far more frequent by chance than others. The empirical determination of background frequencies is necessary to avoid false positives that would merge non-homologous sequences. Conversely, if all sequences are homologous then it is possible to imagine a case where no k-mers are rare. The SARS-CoV-2 genomes are an example of this, as the sequences differ by only a few positions (~3 on average) per thousand. Yet, even in this extreme case, it is possible to identify sequences sharing an unusually high number of rare k-mers. This is sufficient for connecting homologous sequences, which is the goal of the next step in phase 1.

Starting from the first sequence, a depth-first search is commenced to delineate disjoint sets of homologous sequences. That is, any sequences deemed homologous to the first sequence are assigned to a stack and the process is repeated from each of those sequences until no unassigned sequences remain on the stack (Fig. 1B, C). All of the sequences that were visited are assigned to the same partition and share single-linkage homology. Once the stack is empty, the next unassigned sequence becomes the first member of a new partition. This process is repeated until every sequence is assigned a partition (Fig. 1D), thereby visiting every sequence once for a linear time algorithm. In practice, homologous input sequences result in relatively few partitions (e.g., 1 partition) and non-homologous input sequences result in many partitions.

## Phase 2: pre-sorting sequences based on k-mer similarity in linear time

The key innovation underlying the Clusterize algorithm is to pre-sort sequences such that accurate clustering can be performed in linear time. Pre-sorting is conducted within each partition until reaching an iteration limit (i.e., parameter $B$) or the rank ordering of sequences stabilizes. First, a sequence ($j$) is randomly selected from each partition with a probability proportional to its average variability in rank order ($v$) multiplied by its length ($L$): $p_j = v_j * L_j$. This formula encourages the selection of longer sequences that still have not stably sorted. Then, the k-mer distance (1 - k-mer similarity) is calculated between the sequence

and all others, resulting in a vector of distances ($d_1$). Second, another sequence is randomly selected with probability proportional to $d_1$ and inversely proportional to the non-overlapping difference in length ($\Delta ov$) with the first randomly selected sequence: $p_j = d_{1j}/(\Delta ov_j + 1)$. This prioritizes sequences that overlap completely with the first sequence but are dissimilar. Again, the k-mer distance is calculated between the second sequence and all others ($d_2$). The relative difference between these two vectors ($d_1 - d_2$) becomes the initial vector ($D$) upon which the rank order of sequences is determined.

The process of generating new relative distance vectors ($D$) is repeated up to 2000 iterations (i.e., parameter $B$) within each partition. I termed this process relatedness sorting because more similar sequences become closer together with additional iterations ($i$). Only a single vector is kept after each iteration to limit the memory overhead of phase 2. There are multiple ways in which to combine $D_i$ and $D_{i-1}$ into a new unified vector, $D$. I chose to project both vectors onto their axis of maximum shared variance after centering both vectors to have a mean of zero (Fig. 1I). This is equivalent to performing principal component analysis on the two column matrix formed by combining $D_i$ and $D_{i-1}$ and retaining only the first principal component (as $D$). Each vector is ordered with radix sorting to define the rank order of sequences. Clusterize keeps track of the number of positions each sequence moves in the relatedness ordering ($v$) using an exponentially weighted moving average with smoothing parameter α (0.05): $v = \alpha * v_i + (1 - \alpha) * v_{i-1}$. Note that the vector $v_0$ is initialized with values of 2000 (i.e., parameter $C$) and values of $v$ are capped at 2000 in each iteration as an upper limit for practicality.

As relatedness sorting progresses, the rank ordering ($v$) typically stabilizes, with some sequences stabilizing before others. When a sequence's rank order stabilizes below 1000 (i.e., parameter $C/2$), its partition is split into separate groups at this sequence (Fig. 1J), as long as each new group would contain at least 2000 sequences. Relatedness sorting is continued within each group until all of its sequences stabilize in rank order below 1000 or the maximum number of iterations (i.e., parameter $B$) is reached. In practice, most groups stabilize in rank order before 2000 iterations and, thus, parameter $B$ is not a limiting factor. Rank order stability is defined as $v \leq 1000$ because the sequences are moving within 2000 positions on average (i.e., ±1000), which is the number of sequences considered in phase 3 of the algorithm (i.e., parameter $C$).

## Phase 3: clustering sequences in linear time

No sequences were clustered in phases 1 or 2, but the sequences were ordered by relatedness in preparation for clustering. Two linear time strategies are used for clustering sequences: (1) The rare k-mer strategy draws sequences for comparison in the exact same manner as used in phase 1, thereby prioritizing sequences sharing the greatest number of rare k-mers; (2) The relatedness sorting strategy selects proximal sequences from the final rank ordering of sequences determined in phase 2. Up to 2000 (i.e., parameter $C$) total sequences are drawn from the two strategies in proportion to the number of clustered sequences arising from each strategy. That is, when a sequence is added to an existing cluster, Clusterize determines whether the sequence was in the set of sequences originating from rare k-mers and/or relatedness sorting. The count of sequences drawn from each strategy is incremented in accordance with the clustered sequence's origin(s). In this manner, the more lucrative strategy is emphasized when determining where to draw the next 2000 candidate sequences.

K-mer similarity is calculated to the cluster representatives corresponding to all 2000 sequences, and any cluster representative passing the similarity threshold is used for assignment. If none exists, the sequence is aligned with up to 200 (i.e., parameter $E$) cluster representatives having the highest k-mer similarities. If none of these are within the similarity threshold, the sequence becomes a new cluster representative. To avoid issues with partial-length sequences,

the sequences are visited in order of decreasing length so that the cluster representative is always the longest sequence in its cluster.

## Clustering benchmark and program comparison

Clusterize is implemented in the C and R programming languages[18] within the DECIPHER package[17] (v2.30.0) available from Bioconductor[19]. Clusterize's default parameters were used for all tests, while specifying *cutoff* (1 - similarity threshold) and *processors* (number of threads). Five other popular clustering programs were chosen for comparison because they can scale to millions of sequences: CD-HIT/CD-HIT-EST (v4.8.1)[5], ESPRIT-Forest[15], Linclust[16] and MMseqs2 (Release 13-45111; SSE4.1)[6], and UCLUST (USEARCH v11.0.667; 32 bit free version)[7]. Parameters were set to align definitions of minimum sequence coverage across programs and require at least 50% coverage of the target sequence. CD-HIT/CD-HIT-EST required non-default parameters "-M 0", "-aS 0.5", and "-n *word-length*", where *word-length* was decreased as needed from the default of 5 (proteins) or 10 (nucleotides) until the program would run without error. Linclust was run using *mmseqs* command "easy-linclust" with non-default parameters "--cov-mode 1", "-c 0.5" and "--kmer-per-seq 80" to match the higher sensitivity variant of Linclust originally published (i.e., "-m 80"). MMseqs2 was run using the command "easy-cluster" with "--cov-mode 1" and "-c 0.5". UCLUST was run using the *usearch* command "-cluster_fast" with the non-default parameter "-target_cov 0.5". ESPRIT-Forest was run by specifying a single similarity threshold for both the lower ("-l") and upper ("-u") limit. All programs were configured to use 8 processors except when testing scalability with multiple processors.

All runtimes are reported in elapsed time (i.e., wall time). Memory usage is reported as the peak resident set size. Asymptotic scalability was calculated from the slope of a line fit to the highest four measured points of time versus input size in log-log space. Determination of the fraction of sequences exceeding the similarity threshold was performed by randomly sampling up to 10,000 pairs of clustered sequences and calculating their similarity with DECIPHER[17]. Results are only shown when there were at least 5000 clustered sequences. The results shown in Fig. 4 were contrasted with chord diagrams created by the R package *circlize*[48]. AMI and NMI were calculated with the R package *aricode* (v1.0). Label consistency was computed as the average fraction ($f$) of each cluster that shared the cluster's most frequent label, weighted by the size ($m$) of the cluster minus one. In this way, singleton clusters (i.e., $m = 1$), which have 100% label consistency by definition, have no influence on the average consistency. Given $M$ total clusters, average consistency can be formulated as:

$$Consistency = \sum_{i=1}^{M} f_i * (m_i - 1) \Big/ \sum_{i=1}^{M} (m_i - 1)$$

To compare programs for accuracy on homologous sequence sets, I downloaded all proteins from the NCBI (Protein database) matching hidden Markov models in TIGRFAM[26]. A total of 411 protein families had at least 20,000 unique sequences with a labeled genus, and 3001 had fewer than 20,000 unique sequences with a labeled genus. AMI and NMI were computed with the subset of sequences having a class, family, or genus label (typically > 99%). TIGRFAM sequence sets are available from Zenodo (accession 10019584). TIGRFAM tests were performed on Open Science Pool computers matching the minimum specifications: 8 processors, 8 GB of memory, and 10 GB of disk space. For long nucleotide sequences, I downloaded all 13.2 million SARS-CoV-2 genomes available from GISAID (gisaid.org). This set was reduced to the 2 million distinct full-length (29,000 - 30,000 nucleotides) genomes without ambiguities (e.g., "N"). For short nucleotide sequences, I downloaded the set of all quality-trimmed merged paired-end V3-V4 reads collected from healthy human subjects in the Human Microbiome Project. Reads were

labeled by their body site of origin, duplicate reads were removed, and read order was randomized.

To compare programs for accuracy on homologous sequence sets, I downloaded the complete Metaclust benchmark originally published with Linclust[16]. I dereplicated and randomized the order of Metaclust sequences for consistency with the previous benchmarks. To develop another benchmark of non-homologous sequences, I used the set of all bacterial genomes available from RefSeq (release 220). Proteins and their corresponding protein coding sequences were downloaded for each genome, labeled by their protein function predicted by PGAP[49], dereplicated, and hypothetical proteins were discarded. Sequences from randomly selected genomes were appended until reaching the maximum file size of 50 GB allowed by the Zenodo repository at 44,831 genomes. The final sets of matched protein and nucleotide sequences are available from Zenodo (accessions 10030000 and 10031801). For easier reproducibility, all tested subsets were taken starting from the first sequence until reaching the intended number of sequences used for benchmarking. Time and memory analyses were performed on a Dell PowerEdge T650 workstation with two Intel Xeon processors (E5-2690 v4 2.6 GHz) and 792 GB of memory running CentOS 7.

## Reporting summary

Further information on research design is available in the Nature Portfolio Reporting Summary linked to this article.

## Data availability

New sequence sets used for benchmarking were deposited in the Zenodo repository under accessions 10019584, 10030000, and 10031801. All 16S rRNA gene sequences used in this study are available from the Human Microbiome Project Data Portal https://portal.hmpdacc.org, SARS-CoV-2 sequences are available from GISAID, and the Metaclust benchmark is available online https://metaclust.mmseqs.org. GISAID requires an application to access sequence data. Data generated in this study are included in the published article, its Supplementary Information, and the Zenodo repositories listed above.

## Code availability

Clusterize is part of the open source DECIPHER package for the R programming language available from Bioconductor https://doi.org/10.18129/B9.bioc.DECIPHER.

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

## Acknowledgements

This study was funded by the NIAID at the NIH (grant number 1U01AI176418-01 to ESW). Computer resources for this study were provided by the Open Science Grid. I acknowledge the data contributors to the GISAID initiative for generating the SARS-CoV-2 sequences used in this study, and I am grateful to the Human Microbiome Project for making their datasets accessible.

## Author contributions

E.S.W. is the sole author.

## Competing interests

The author declares no competing interests.
