## [Peer Review File · Nature Communications]

Accurately clustering biological sequences in linear time by relatedness sortingReviewer #1 (Remarks to the Author):

The paper describes novel method, Clusterize, for clustering sets of nucleotide and protein sequences by similarity. Importantly, the method's run times scales linearly with the number of sequences. On the presented protein benchmark test, it shows better sensitivity than CD-HIT and another linear-time algorithm, Linclust (fig 2). On two nucleotide datasets (fig 3), it shows better sensitivity than mmseqs, uclust, and linclust, and somewhat lower sensitivity than CD-HIT, although CD-HIT constructed wider clusters. On the other hand, it is among the slower methods on this dataset and is about 3 times slower than CD-HIT.

Main points

1. The method is only tested on sets of homologous sequences. That is a special case of sequence clustering, in which usually sets are much smaller and there is much less need for very fast method. Also, specialized tools are available for this purpose, such as usearch -cluster_otus, FastANI, , which Clusterize should be compared with.

The linear-time pre-sorting step in Clusterize is unlikely to work if sequences are not homologous, as acknowledged in the paper. To run on general sequence set containing non-homologous sequences, a preclustering with k-means is required, but this algorithm has not been benchmarked.

2. The method is advertised to be able to cluster huge datasets, but the largest of the test sets has only 16 million quite short RNA sequences. Many datasets from metagenomics consist of billions to tens of billions of sequences. For instance, Linclust clustered 1.6 billion protein sequences of similar length in their 2017 paper. The current benchmarks do not show a decisive speed advantage over existing tools. For that purpose, a benchmark on huge sequence sets (>1 billion sequences) of protein and nucleotide sequences is needed. Also, diverse clustering thresholds should be shown, not only 99.99% and 99% as in fig 3.

3. Benchmarks on both protein and nucleotide sequences should include a run time analysis, a run time versus number of core analysis, and a memory analysis. It is worrying that Clusterize fails to cluster 3 million virus sequences due to insufficient memory on 256 GB of main memory. It is written that linclust failed to cluster 2 million sequences due to insufficient memory but Linclust and mmseqs have been shown to cluster much larger datasets in memory and also have an option to cluster datasets that do not fit into main memory.

Minor points

4. The protein sequence sets for fig 2 are all small (up to 100k sequences). It is also unclear why only 411 out of the 4488 protein family sets in TIGRFAM were selected, and according to which criteria.

5. Relatedness sorting is clustering in one dimension. It is unclear how this can be selective enough for large datasets, since many dissimilar sequences should fall near the same point when projected from a high-dimensional space onto a one-dimensional line.

6. ``A random sample of 10,000 pairs of clustered sequences was drawn from each program's clusters" At which numberof sequences was this done?

Reviewer #2 (Remarks to the Author):

Thanks for the opportunity to review this paper. The manuscript by Dr Erik Wright presents a new sequence clustering method (Clusterize) providing linear time complexity while maintaining similar accuracy than less scalable approaches.

Clustering large amounts of sequences based on their similarity is a computationally complex problem and, clearly, a limiting step for many comparative genomic analyses. Currently, MMseqs2's iterative searches (combined with the Linclust algorithm) is probably one of the best options balancing speed and accuracy; an approach that has been already adopted by many metagenomic surveys and reference genomic databases.

Here, Dr. Wright shows that the Clusterize algorithm achieves the same accuracy as MMseqs2 and UClust (Figure 2), while keeping better (linear) scalability (Figure 3). However, although this is of course promising, I would suggest that further analyses are provided to make sure that the Clusterize improvements, compared to current methods, are not marginal nor confined to very specific scenarios.

For instance, I am particularly concerned about the fact that accuracy and scalability comparisons were performed on very different datasets and under very different similarity thresholds.

On one side, clustering accuracy was measured using only 411 TIGRFAM protein families as a reference, and under identity thresholds ranging from 50% to 95%. On the other side, scalability was assessed on two much larger sets of nucleotide sequences (instead of proteins) and at highly stringent identity thresholds (99.99% for SARS-2 genomes and 99% for 16S genes).

Therefore, doubts remain whether 1) linear scaling is actually maintained when applied on large protein sets with lower identity thresholds, and 2) performance improvements compared to currently available methods are still significant under those thresholds. Note that in the recent Linclust paper (PMID: 29959318), authors reported linear scaling for clustering sequences at 90% identity using both MMseqs2 and Linclust: "At 90% sequence identity threshold (...) runtimes scale very roughly quadratically for UCLUST (N1.62) and CD-HIT (N2.75) whereas they grow only linearly for Linclust/MMseqs2 (N0.94) and Linclust (N1.01)."

If that is the case, Clusterize performance improvements compared to the MMseqs2 approach might be only relevant when working at ultra-stringent identity thresholds such as the ones used in Figure 3 (e.g. 99.99% nucleotide identity). It would be great that this potential inconsistency is clarified with additional tests. In short: are Clusterize improvements dependent on the identity threshold, sequence type or input size?

In addition, to facilitate comparisons, it would be nice to measure Clusterize performance on the very same dataset and benchmark used in the Linclust paper, where sequences per cluster and runtimes were also measured for a large protein set at different identity thresholds. Moreover, a direct comparison against the iterative MMseqs2+Linclust approach (which is probably the method of choice for most current studies) under different thresholds, data sizes and program parameters would be necessary to illustrate more clearly that Clusterize improvements will indeed have a significant impact (either on runtimes or accuracy) on very large-scale surveys (e.g. recent metagenomics studies can easily involve hundreds of millions of sequences).

On a different note, I am personally curious about the ability of Clusterize to identify clusters at very low identity thresholds (e.g. below 50% protein identity). Regardless of whether those clusters could or could not be considered protein families (i.e. beyond the twilight zone), it might be useful to know the technical limitations of the method.

Regarding clustering accuracy measurements, I personally think the TIGRFAM benchmark used for Figure 2 (which is based on 411 prokaryotic protein families and taxonomy-based AMI/NNI scores) represents a rather small and potentially biased comparison. Note that: 1) the relationship between protein identity thresholds (50-95%) and the taxonomic ranks used for benchmarking is in principle unknown (clusters at 50% are probably expected to group sequences from higher ranks such as class or phylum). 2) Rank assignments in the NCBI taxonomy are not necessarily consistent with the actual evolution of species (PMID: 30148503), which would interfere with the labelling and scoring system. And 3) horizontal gene transfer is common in Bacteria and Archaea. If present in this benchmark set, it would obviously affect the scoring system.

For such a small reference set (411 families, ~20,000 unique proteins), my suggestion would be to run an exact clustering algorithm at different identity thresholds. That should provide a better ground truth to compare with and would eliminate the need of potentially misleading scores based on taxonomic interpretations. I would also suggest expanding the number of families tested (from the methods, it was not clear to me why the benchmark was limited to 411 families, when TIGRFAM seems to provide thousands).

Overall, I think Clusterize represents a promising and very interesting method if performance differences compared to current approaches are more exhaustively validated and confirmed.

I thank the reviewers for their comments on the first manuscript draft. Here, I provide a major revision of the manuscript to address the reviewers' concerns. Highlights include:

(1) Both reviewers questioned whether the Clusterize algorithm would perform less well on inputs containing non-homologous sequences. They also desired to see performance on even larger input sets. To address this comment, I generated an enormous benchmark set with high quality annotations, which required about 4 million hours of grid time to construct. The benchmark set contains annotated protein coding sequences from one million bacterial genomes. Also, I downloaded the set of unique intergenic regions from all genomes. Both of these benchmarks contain 32 million unique sequences. I hope the two new benchmarks are amply large and diverse to assuage any previous reviewer concerns.

(2) The reviewers were skeptical that the initial k-means partitioning algorithm would be sufficient for non-homologous sequence inputs. In response, I modified the first and last phases of the Clusterize algorithm to better perform on diverse sets of input sequences. These modifications maintained the linear time complexity of the original algorithm and maintained its performance on the original manuscript benchmarks. I revised the text accordingly, and there are new panels in Figure 1 depicting the algorithm modifications. I hope these changes better explain how Clusterize can achieve high accuracy on highly diverse sequence sets. I thank the reviewers for encouraging me to further improve the Clusterize algorithm.

(3) The first reviewer requested addition of OTU specific clustering programs and the second reviewer suggested comparison to hierarchical (exact) clustering. I addressed these comments by adding an additional program, ESPRIT-Forest, that was designed to perform hierarchical (exact) clustering of 16S reads in sub-quadratic time. ESPRIT-Forest is a parallel implementation of ESPRIT-Tree, which was previously shown to outperform OTU clustering approaches in accuracy. Unfortunately, it was limited to the 16S benchmark because it is nucleotide specific and quickly ran out of memory on the other nucleotide benchmarks. Nevertheless, the addition of ESPRIT-Forest highlights the comparative lack of scalability of exact algorithms.

REVIEWER COMMENTS

Reviewer #1 (Remarks to the Author):

The paper describes novel method, Clusterize, for clustering sets of nucleotide and protein sequences by similarity. Importantly, the method's run times scales linearly with the number of sequences. On the presented protein benchmark test, it shows better sensitivity than CD-HIT and another linear-time algorithm, Linclust (fig 2). On two nucleotide datasets (fig 3), it shows better sensitivity than mmseqs, uclust, and linclust, and somewhat lower sensitivity than CD-HIT, although CD-HIT constructed wider clusters. On the other hand, it is among the slower methods on this dataset and is about 3 times slower than CD-HIT.

I thank the reviewer for the constructive comments.

Main points

1. The method is only tested on sets of homologous sequences. That is a special case of sequence clustering, in which usually sets are much smaller and there is much less need for very fast method. Also, specialized tools are available for this purpose, such as `usearch -cluster_otus`, `FastANI`, , which `Clusterize` should be compared with.

In response to this comment, I added a test of non-homologous sequences derived from the proteome of all publicly available bacterial genomes. In order to be able to gauge accuracy, I restricted the set of all bacterial proteins to those that were annotated with IDTAXA using the KEGG database. In summary, protein sequences were annotated and these annotations were transferred to their corresponding nucleotide sequences to create an immensely diverse sequence set with accurate annotations. Clustering was performed at four similarity thresholds. This benchmark showed that `Clusterize` is accurate and scalable.

I also added a new benchmark composed of all unique intergenic regions found in the same set of bacterial genomes. This set is extremely diverse but lacks the annotations required for quantifying accuracy. Nevertheless, the results on this new benchmark further confirm the scalability of `Clusterize`, which also produced the fewest clusters of any program.

I looked into the two specialized tools the reviewer mentioned. `FastANI` is designed to rapidly perform all-v-all comparisons of two genomes, but it does not mention clustering as an application as far as I can tell. `USEARCH cluster_otus` is designed to perform 97% similarity OTU clustering of amplicon reads. However, the developer suggests upgrading to `UPARSE-OTU`, which is designed to perform simultaneous chimera removal and clustering at a *fixed* 97% similarity threshold. I don't believe this program is applicable to the previous benchmark of clustering 16S reads at 99% similarity. I found fixing the similarity threshold at 97% to be strange because the same developer also has a paper stating the 97% threshold is too permissive for microbiome work, which I had already cited in the first draft of the manuscript.

Nevertheless, the focus of this study is to benchmark against general purpose clustering algorithms rather than those limited to a specific application, such as amplicon sequencing workflows with chimera removal. The amplicon clustering problem is specialized in that the goal is to cluster sequence errors within a single sample, which permits statistical models of substitution rates (in the manner of `DADA2` ASV clustering) that collapse some substitutions more than others. The amplicon sequencing benchmark data used in this study was already processed by sample, and the goal was instead to cluster across samples. This is a distinct objective to that of `UPARSE` and other OTU clustering algorithms that seek to denoise amplicon sequencing data.

Therefore, to address the reviewer's concern, I added another program that was designed for clustering homologous sequences: `ESPRIT-Forest`. It is a parallelized implementation of `ESPRIT-Tree`, which was shown to outperform inexact clustering algorithms and is widely used. Unfortunately, its use was limited to the 16S benchmark because it is limited to nucleotide

sequences and immediately ran out of memory on non-homologous sequences and SARS-CoV-2 genomes. Nevertheless, the addition of this program broadened the clustering comparisons and I thank the reviewer for this suggestion.

The linear-time pre-sorting step in Clusterize is unlikely to work if sequences are not homologous, as acknowledged in the paper. To run on general sequence set containing non-homologous sequences, a preclustering with k-means is required, but this algorithm has not been benchmarked.

This is an important point, and one that is addressed with the new benchmark containing very diverse sequences. I thank the reviewer for pointing this out.

2. The method is advertised to be able to cluster huge datasets, but the largest of the test sets has only 16 million quite short RNA sequences. Many datasets from metagenomics consist of billions to tens of billions of sequences. For instance, Linclust clustered 1.6 billion protein sequences of similar length in their 2017 paper. The current benchmarks do not show a decisive speed advantage over existing tools. For that purpose, a benchmark on huge sequence sets (>1 billion sequences) of protein and nucleotide sequences is needed. Also, diverse clustering thresholds should be shown, not only 99.99% and 99% as in fig 3.

The new benchmark is similar to the original Linclust benchmark in that it encompasses an enormous diversity and scale of sequences. However, the new benchmark uses IDTAXA annotations with KEGG groups rather than the GO terms, which have been previously shown to be unreliable because they are too general to benchmark accuracy (PMIDs: 21364756, 21551147, & 22479173). To address the reviewer's points, I benchmarked all clustering programs at a wide variety of similarity thresholds with increasing numbers of nucleotide sequences until the point where they ran out of time or memory. This is shown in a new results section and associated figure in the revised manuscript. I hope the reviewer is satisfied the additional analyses further confirm the merits of Clusterize.

Clusterize could theoretically scale past 2 billion input sequences given sufficient time and memory. If 20 million sequences requires about a day, 200 million (~3 weeks) is imaginable, but 2 billion (~100 days) is not. Linclust has a significant advantage in the regime beyond 200 million distinct input sequences. However, given the poor accuracy of Linclust on all the benchmark tests in this manuscript, I am skeptical that Linclust is sufficient for clustering even low numbers of sequences. I do not know of an existing solution for accurately clustering billions of sequences, as the reviewer requests. Clusterize is currently not of practical utility for this purpose. In my view, clustering billions of distinct sequences is an interesting open problem, but it is not the focus of the current manuscript.

3. Benchmarks on both protein and nucleotide sequences should include a run time analysis, a run time versus number of core analysis, and a memory analysis. It is worrying that Clusterize fails to cluster 3 million virus sequences due to insufficient memory on 256 GB of main memory. It is written that linclust failed to cluster 2 million sequences due to insufficient

memory but Linclust and mmseqs have been shown to cluster much larger datasets in memory and also have an option to cluster datasets that do not fit into main memory.

I investigated and fixed the previously reported memory limitation with Clusterize. I now show how Clusterize manages to cluster the full set of SARS-CoV-2 genomes. It is unclear to me why Linclust reported an out of memory error when clustering the maximum number of SARS-CoV-2 genomes, but I added more memory to the benchmarking server so that this was no longer a concern. I also added a scalability analysis with for memory and the numbers of cores, as the reviewer requested. With default settings, Clusterize uses about 3-fold more memory than Linclust and MMseqs2. The results are shown in new figure panels in the revised manuscript.

I also added a runtime analysis versus the number of processors, as the reviewer requested. The results are provided in a new figure. This analysis led me to discover that UCLUST does not return the same results with different numbers of processors, which is also shown in the new figure.

Minor points

4. The protein sequence sets for fig 2 are all small (up to 100k sequences). It is also unclear why only 411 out of the 4488 protein family sets in TIGRFAM were selected, and according to which criteria.

This information was already included in the previous version of the manuscript and, therefore, no changes were made in response to this comment:

"A total of 411 protein families had at least 20,000 unique sequences (20,010 to 108,425) labeled with family and genus taxonomic ranks, with average lengths between 70 and 1,562 amino acids."

5. Relatedness sorting is clustering in one dimension. It is unclear how this can be selective enough for large datasets, since many dissimilar sequences should fall near the same point when projected from a high-dimensional space onto a one-dimensional line.

I agree with the reviewer that it is impressive how well relatedness sorting works. When I originally came up with the idea, I was unsure it would work until I implemented it and it worked well in tests. I think the analogy of the phylogenetic tree is insightful, because any number of sequences can be arranged in a line, even if they may share no detectable homology. Long branches may lead down to distinct groups of sequences, such that neighboring sequences sometimes share no homology, but sequences are generally adjacent to their nearest neighbor. The number line is infinite, so hypothetically all sequences could be arranged in this manner. Effectively, this approach leverages the fact that biological sequences are related. If random sequences were input then it would not work, but clustering is also meaningless in such a case.

6. "A random sample of 10,000 pairs of clustered sequences was drawn from each program's clusters" At which number of sequences was this done?

This has been clarified in the manuscript. I now show plots of sequences below the user-specified similarity threshold at each number of input sequences.

Reviewer #2 (Remarks to the Author):

Thanks for the opportunity to review this paper. The manuscript by Dr Erik Wright presents a new sequence clustering method (Clusterize) providing linear time complexity while maintaining similar accuracy than less scalable approaches.

Clustering large amounts of sequences based on their similarity is a computationally complex problem and, clearly, a limiting step for many comparative genomic analyses. Currently, MMseqs2's iterative searches (combined with the Linclust algorithm) is probably one of the best options balancing speed and accuracy; an approach that has been already adopted by many metagenomic surveys and reference genomic databases.

Here, Dr. Wright shows that the Clusterize algorithm achieves the same accuracy as MMseqs2 and UClust (Figure 2), while keeping better (linear) scalability (Figure 3). However, although this is of course promising, I would suggest that further analyses are provided to make sure that the Clusterize improvements, compared to current methods, are not marginal nor confined to very specific scenarios.

I thank the reviewer for their feedback on the initial manuscript draft.

For instance, I am particularly concerned about the fact that accuracy and scalability comparisons were performed on very different datasets and under very different similarity thresholds.

On one side, clustering accuracy was measured using only 411 TIGRFAM protein families as a reference, and under identity thresholds ranging from 50% to 95%. On the other side, scalability was assessed on two much larger sets of nucleotide sequences (instead of proteins) and at highly stringent identity thresholds (99.99% for SARS-2 genomes and 99% for 16S genes).

Therefore, doubts remain whether 1) linear scaling is actually maintained when applied on large protein sets with lower identity thresholds, and 2) performance improvements compared to currently available methods are still significant under those thresholds. Note that in the recent Linclust paper (PMID: 29959318), authors reported linear scaling for clustering sequences at 90% identity using both MMseqs2 and Linclust: "At 90% sequence identity threshold (...) runtimes scale very roughly quadratically for UCLUST (N1.62) and CD-HIT (N2.75) whereas they grow only linearly for Linclust/MMseqs2 (N0.94) and Linclust (N1.01). "

If that is the case, Clusterize performance improvements compared to the MMseqs2 approach might be only relevant when working at ultra-stringent identity thresholds such as the ones used in Figure 3 (e.g. 99.99% nucleotide identity). It would be great that this potential inconsistency is clarified with additional tests. In short: are Clusterize improvements dependent on the identity threshold, sequence type or input size?

In addition, to facilitate comparisons, it would be nice to measure Clusterize performance on the very same dataset and benchmark used in the Linclust paper, where sequences per cluster and runtimes were also measured for a large protein set at different identity thresholds. Moreover, a direct comparison against the iterative MMseqs2+Linclust approach (which is probably the method of choice for most current studies) under different thresholds, data sizes and program parameters would be necessary to illustrate more clearly that Clusterize improvements will indeed have a significant impact (either on runtimes or accuracy) on very large-scale surveys (e.g. recent metagenomics studies can easily involve hundreds of millions of sequences).

In the revised manuscript, I added two large benchmarks of non-homologous sequences. I hope these new analyses are sufficient to assuage any reviewer concerns. I chose not to use the original Linclust dataset because it was annotated with GO terms, which are overly general for the purposes of benchmarking accuracy (PMIDs: 21364756, 21551147, & 22479173). The new benchmark sets are enormous, diverse, and accurately annotated. I think these new analyses provide a fair comparison across a wide diversity of inputs and similarity cutoffs.

Benchmarking is a challenge for comparing clustering programs. The 411 sets of TIGRFAM protein families with 20,000 or more sequences is already one of the most considerable datasets in size and scope. The two nucleotide benchmarks were selected to provide realistic user scenarios. The SARS-CoV-2 genomes are nearly identical, which requires a high similarity threshold (near 100%). Similarly, 16S amplicons datasets are typically clustered at very high similarity thresholds ($\geq 97\%$). I believe the full set of benchmarks now adequately spans the variety of suitable user scenarios.

On a different note, I am personally curious about the ability of Clusterize to identify clusters at very low identity thresholds (e.g. below 50% protein identity). Regardless of whether those clusters could or could not be considered protein families (i.e. beyond the twilight zone), it might be useful to know the technical limitations of the method.

I extended the clustering from 50% to 40% identity to address this comment. As shown in Figure 2, Clusterize is competitive at this lower-identity range, indicating its performance is maintained at low similarity.

Regarding clustering accuracy measurements, I personally think the TIGRFAM benchmark used for Figure 2 (which is based on 411 prokaryotic protein families and taxonomy-based AMI/NNI scores) represents a rather small and potentially biased comparison. Note that: 1) the relationship between protein identity thresholds (50-95%) and the taxonomic ranks used for

benchmarking is in principle unknown (clusters at 50% are probably expected to group sequences from higher ranks such as class or phylum). 2) Rank assignments in the NCBI taxonomy are not necessarily consistent with the actual evolution of species (PMID: 30148503), which would interfere with the labelling and scoring system. And 3) horizontal gene transfer is common in Bacteria and Archaea. If present in this benchmark set, it would obviously affect the scoring system.

For such a small reference set (411 families, ~20,000 unique proteins), my suggestion would be to run an exact clustering algorithm at different identity thresholds. That should provide a better ground truth to compare with and would eliminate the need of potentially misleading scores based on taxonomic interpretations. I would also suggest expanding the number of families tested (from the methods, it was not clear to me why the benchmark was limited to 411 families, when TIGRFAM seems to provide thousands).

I understand the reviewer's concerns about the TIGRFAM benchmark. First, there are only 411 TIGRFAM families with at least 20,000 sequences. Second, it was noted in the initial manuscript draft that taxonomy is an imperfect proxy for clustering. I considered the reviewer's suggestion to use exact (hierarchical) clustering to define groups. However, most of the families have far more than 20k sequences and it is impractical to hierarchically cluster this many sequences. I attempted to use ESPRIT-Forest, an exact parallelized hierarchical clustering algorithm, but it immediately failed on non-homologous sequences. Therefore, hierarchical clustering is not a reasonable alternative to taxonomy for benchmarking accuracy. Third, I added the class rank to this plot to address the reviewer's concern that low identity thresholds are insufficiently represented by genus and family ranks. The phylum rank was not reasonable to use, as their becomes an extreme imbalance in group sizes at this rank. Fourth, the new protein coding sequences benchmark is based on IDTAXA annotations using the KEGG database. These annotations are highly accurate and provide an alternative to taxonomy as the biological ground truth. These two independent labeling approaches (i.e., taxonomy and function) both yielded similar conclusions in terms of how the different programs compare across benchmarks.

Overall, I think Clusterize represents a promising and very interesting method if performance differences compared to current approaches are more exhaustively validated and confirmed.

Thank you again for the constructive comments.

Reviewer #1 (Remarks to the Author):

The chief limitation pointed out previously has been addressed well: the method is now able to cluster also sequence sets containing non-homologous groups. A preclustering step (phase 1) has been added for this purpose. Also, benchmarks containing groups of non-homologous sequences have been added, in particular one consisting of 32 million coding nucleotide sequences. These changes have strengthened the manuscript considerably. The manuscript seems to introduce a number of interesting original ideas, however, it is very difficult to judge their merits due to the very vague and incomplete description of the methods.

The overall performance in terms of sensitivity and speed in comparison with MMseqs, Linclust, Uclust and CDHIT is good. On the 32M sequence benchmark (fig 6), it is about 10x faster than Uclust and CDHIT at similar clustering quality and more sensitive than Linclust at ~30x slower speed. It is slower than MMseqs and as accurate, except at 100% sequence id, where it is significantly more accurate.

Major issues:

1. The entire description of the method is *very* intransparent because most of the important information is missing that is would be required if someone were to reimplement this method. Not even the broad outlines are sufficiently clear. The discription in the Results does not help much in this regards, nor does the supposedly more detailed description in the Methods section. This will make it hard for the reader to judge the methodological advances and appreciate which ideas are crucial and which are not important, and to learn from and build on the key ideas.

Here is an incomplete list of examples.

- * Line 96, line 598: "First, a sequence is randomly selected and up to 100 k-mers are drawn." Up to 100? How do you exactly compute the number of k-mers to be drawn?
- * Line 98,603-604: How do you calculate the p-value for the binomial test given that you have, say, l_1 and l_2 k-mers selected in sequences 1 and 2, respectively, and l_{12} are shared between them?
- * Line 99: "Sequences are assigned to the group using a single bit per sequence..." I cannot even guess what is meant. Describe the data structure that is used to hold that information and how it is used.
- * Lines 100-102: Do you reassign sequences later if they turn out to have a better p-value with a cluster center found at a later stage of the clustering?
- * Lines 102-105: "The resulting bit vectors specifying group membership are used to construct a matrix with the p-value that the overlap in group membership is statistically significant using a binomial test." Unclear. Please explain in detail.
- * Lines 599-600: "Clusterize uses a simple hash function that counts the number of integer encoded k-mers falling into a fixed set of equal width bins." This is not clear. Write down the formula for the hash function explicitly.
- * How do you ensure that the k-mers do not overlap by k-1 positions, since if one is rare the neighboring k-mers shifted by +/- 1 will also be rare?
- * Line 609: "A p-value matrix is calculated from the overlap between group membership using a binomial test." Unclear. Please write down the formula for the p-value explicitly.
- * Is k-mer anchoring similarity (Fig 1M) computed everywhere that a k-mer distance is mentioned in the methods? For example in phase 1? phase2? phase3?
- * Lines 651-653: "Second, another sequence is randomly selected with probability proportional to d_1 and inversely proportional to the non-overlapping difference in length with the first sequence." Unclear. Write down the exact formula for the probability.
- * Lines 669-670: "For each sequence, the subset of (up to 65,536) sequences sharing at least one rare k-mer is selected." How do you do that exactly? What data structure do you use? How do you

make this critical part efficient? For instance, to count the number of k-mers a sequence has in common with all other sequences in the database, an inverted k-mer index is usually built. Is that what you do? What exactly does this index look like? Is it the main driver for the memory footprint? Why is this part $O(N)$? And why and how do you cut off at 65536?

* Lines 673-674: "Up to C total sequences are drawn from the two strategies in proportion to how many clusters arose from each strategy." Unclear. How is that done exactly? How can we know how many clusters arose from each strategy *during* the clustering? And how is C chosen robustly to make sure that it is not too low for some datasets?

* Lines 698-700: "k is set such that only 10% of the selected (up to 65,536) k-mers are expected to be due to chance." Unclear. How is the k-mer length set exactly? Give the formula and explain it.

* Same lines: if the probability for a random k-mer match with any of the input is only 0.1, then the probability to have more than 2 random matches with a specific non-homologous sequence will be very low, won't it? So why do you have to count the number of k-mer matches at all instead of just demanding at least two matches?

* How does the k-mer anchoring similarity algorithm work? What makes it fast? How is the similarity defined? Total number of compatible k-mers? Or number per overlap?

* Line 652: "Second, another sequences is randomly selected with probability proportional to d1 and inversely proportional to the non-overlapping difference in length with the first sequence." It is unclear what is calculated here. Please write down the probability explicitly as mathematical formula.

* Analysis for figures 2, 3, and 6: How were input set sizes reduced? By randomly sampling subsets of defined sizes?

* Are alignments between amino acid sequences computed using a standard substitution matrix?

2. Benchmarking.

a) Please include all TIGRFAM families in the input set. Explicitly excluding smaller families seems quite artificial.

b) The only analysis of protein sequence clustering is shown Fig. 2, but it does not include an analysis of runtime and memory. Either inclusion in the analysis of Fig 3 or at least the runtimes on the full set for the 5 tools would be helpful, to see if Clusterize is competitive with the other tools.

c) How does Clusterize perform with only the k-mer strategy in phase and with only the relatedness sorting in phase 3?

d) Clusterize chooses 100 k-mers per sequence in phase 1 and 50 in phase 3. Linclust chooses by default only 20, and this can be changed by -m option. In their 2018 paper, linclust authors showed results for both -m20 and -m80 and -m80 had much better performance than the default -m20 setting at only small runtime cost. So Linclust -m80 should be included in the Clusterize benchmarks (or replace the -m20 version).

e) I suspect the minimum coverage between sequences to be a strong hidden confounder in these benchmarks, as this setting is expected to have a much larger influence on the clustering quality than the sequence identity and the actual clustering algorithm. For instance, in Figure 6, at 100% sequence identity threshold, MMseqs and LIInclust cluster much fewer sequences together than Clusterize and CDHIT, even though it is obviously trivial to find the 100% identical sequence pairs. The only explanation I can think of for this large divergence is different minimum coverage settings in the tools. Please check this and choose the same settings for all tools to ensure comparability across tools in all benchmarks.

3. What values are set for the parameters A,B,C,D,E set in the benchmarks? How were these values selected? Do any of these parameters depend on the input dataset? If yes, how?

Reviewer #2 (Remarks to the Author):

I thank the author for the comments and additional work. Unfortunately, I could not say that my original concerns are dispelled with the revised version of the manuscript.

- I suggested using the very same dataset as the one used in the Linclust paper because I thought that this would be the best way of directly comparing approaches. Even if GO terms are not the best functional choice for benchmarking, such a comparison would have provided an orthogonal and straightforward validation of Clusterize, which I think would be one of the main questions that any potential user of Clusterize would like to answer: Can I cluster better and faster than Linclust/MMseqs? Even if ignoring GO terms, it would be possible to compare runtimes, number of clusters produced and other technical parameters.

- Dr. Wright argues that the new datasets added are enormous, but I do not agree. Linclust was tested on 1.6 billion metagenomic sequences. The newly added intergenic regions dataset is 32 million. So, the question remains: Is there any limitation on the size of the datasets that Clusterize can handle? I don't think that the performance and scalability increase claimed in the paper can be fully supported without a formal comparison based on datasets of at least a similar size range.

- I also suggested improving/expanding the TIGRFAM benchmark, as it was very small (only 411 protein families) and perhaps biased due to various confounders. As an alternative, I suggested comparing Clusterize results with the exact clustering algorithms, just to confirm that results were accurate from a technical point of view. Unfortunately, none of the options were possible. Generating ground-truth results based on exact-clustering algorithms was argued to be impractical and therefore not addressed. However, I still think that, for such as a small set of sequences, other controls could have been used. 20,000 sequences are in principle addressable by hierarchical clustering algorithms or orthology clustering methods.

- Also, the new panel in Figure 2 showing benchmarking results at the class level does not solve the potential biases. Actually, it seems to suggest that CD-HIT and MMSeqs are performing better (higher peaks) than Clusterize.

- Most importantly, even if the accuracy of all methods are considered very similar (as Figure 2 seems to suggest), the only substantial benefit of Clusterize would reside in runtimes and scalability improvements, which, as I mentioned before, I don't think it has been fully proven yet.

I thank the reviewers for their insightful feedback on the first manuscript revision. The new version addresses their comments. In summary:

(1) The first reviewer requested many methods clarifications. The Methods section was rewritten to address this comment. Also, the Clusterize algorithm was revised to be simpler and, therefore, easier to understand. Parameter optimization is now shown in a supplemental figure (new Fig. S1).

(2) Both reviewers desired more benchmarking. The new manuscript includes results on the Metaclust benchmark (Fig. 3) originally from the Linclust paper and results on an expanded set of TIGRFAM protein families (new Fig. S3). The final benchmark in the manuscript, which previously included coding sequences from genomes, was modified to include both nucleotide and protein sequences. It now closely mirrors the Uniprot benchmark originally published in the Linclust paper but includes coding (nucleotide) sequences, which would not be possible using Uniprot.

(3) Reviewer 1 recommended unifying sequence coverage across programs. Therefore, all benchmark results were re-evaluated under new program parameterization to match coverage. Also, reviewer 1 suggested benchmarking with Linclust set to `-m 80`, because this argument increases sensitivity. I replaced all Linclust results with `--kmer-per-seq 80`. Linclust showed improved sensitivity, although still noticeably lower than Clusterize.

(4) Reviewer 1 asked for a comparison of relatedness sorting and rare k-mers. Clusterize reports how many clustered sequences originate from each approach and there is now a figure showing these results across similarity thresholds (new Fig. S2). It shows that both approaches are highly effective, but relatedness sorting dominates at lower similarity thresholds and rare k-mers at higher similarity thresholds.

Reviewer #1 (Remarks to the Author):

The chief limitation pointed out previously has been addressed well: the method is now able to cluster also sequence sets containing non-homologous groups. A preclustering step (phase 1) has been added for this purpose. Also, benchmarks containing groups of non-homologous sequences have been added, in particular one consisting of 32 million coding nucleotide sequences. These changes have strengthened the manuscript considerably. The manuscript seems to introduce a number of interesting original ideas, however, it is very difficult to judge their merits due to the very vague and incomplete description of the methods.

The overall performance in terms of sensitivity and speed in comparison with MMseqs, Linclust, Uclust and CDHIT is good. On the 32M sequence benchmark (fig 6), it is about 10x faster than Uclust and CDHIT at similar clustering quality and more sensitive than Linclust at ~30x slower speed. It is slower than MMseqs and as accurate, except at 100% sequence id, where it is significantly more accurate.

I thank the reviewer for their encouraging comments.

Major issues:

1. The entire description of the method is *very* intransparent because most of the important information is missing that is would be required if someone were to reimplement this method. Not even the broad outlines are sufficiently clear. The discription in the Results does not help much in this regards, nor does the supposedly more detailed description in the Methods section. This will make it hard for the reader to judge the methodological advances and appreciate which ideas are crucial and which are not important, and to learn from and build on the key ideas.

I greatly appreciate the reviewer's substantial effort to help me improve the manuscript. The Methods section has been completely rewritten in this revision. It should address all of the points made by the reviewer above and many more.

Here is an incomplete list of examples.

* Line 96, line 598: "First, a sequence is randomly selected and up to 100 k-mers are drawn." Up to 100? How do you exactly compute the number of k-mers to be drawn?

No longer needed in the new phase 1.

* Line 98,603-604: How do you calculate the p-value for the binomial test given that you have, say, l_1 and l_2 k-mers selected in sequences 1 and 2, respectively, and l_{12} are shared between them?

No longer needed in the new phase 1.

* Line 99: "Sequences are assigned to the group using a single bit per sequence..." I cannot even guess what is meant. Describe the data structure that is used to hold that information and how it is used.

No longer needed in the new phase 1.

* Lines 100-102: Do you reassign sequences later if they turn out to have a better p-value with a cluster center found at a later stage of the clustering?

All assignment of cluster numbers occurs only in phase 3. This should now be clear in the revised Methods.

* Lines 102-105: "The resulting bit vectors specifying group membership are used to construct a matrix with the p-value that the overlap in group membership is statistically significant using a binomial test." Unclear. Please explain in detail.

No longer needed in the new phase 1.

* Lines 599-600: "Clusterize uses a simple hash function that counts the number of integer encoded k-mers falling into a fixed set of equal width bins." This is not clear. Write down the formula for the hash function explicitly.

The hash function is now described in detail in the revised Methods.

* How do you ensure that the k-mers do not overlap by k-1 positions, since if one is rare the neighboring k-mers shifted by +/- 1 will also be rare?

This should now be clear in the revised Methods.

* Line 609: "A p-value matrix is calculated from the overlap between group membership using a binomial test." Unclear. Please write down the formula for the p-value explicitly.

No longer needed in the new phase 1.

* Is k-mer anchoring similarity (Fig 1M) computed everywhere that a k-mer distance is mentioned in the methods? For example in phase 1? phase2? phase3?

This is now explicitly stated in the revised Methods.

* Lines 651-653: "Second, another sequence is randomly selected with probability proportional to d_1 and inversely proportional to the non-overlapping difference in length with the first sequence." Unclear. Write down the exact formula for the probability.

The formula is provided and described in the revised Methods.

* Lines 669-670: "For each sequence, the subset of (up to 65,536) sequences sharing at least one rare k-mer is selected." How do you do that exactly? What data structure do you use? How do you make this critical part efficient? For instance, to count the number of k-mers a sequence has in common with all other sequences in the database, an inverted k-mer index is usually built. Is that what you do? What exactly does this index look like? Is it the main driver for the memory footprint? Why is this part $O(N)$? And why and how do you cut off at 65536?

This part of the algorithm is detailed in the revised Methods subsection for phase 1 and noted again in phase 3.

* Lines 673-674: "Up to C total sequences are drawn from the two strategies in proportion to how many clusters arose from each strategy." Unclear. How is that done exactly? How can we know how many clusters arose from each strategy *during* the clustering? And how is C chosen robustly to make sure that it is not too low for some datasets?

This should now be clear in the revised Methods.

* Lines 698-700: "k is set such that only 10% of the selected (up to 65,536) k-mers are expected to be due to chance." Unclear. How is the k-mer length set exactly? Give the formula and explain it.

Formulas for choosing k are now provided.

* Same lines: if the probability for a random k-mer match with any of the input is only 0.1, then the probability to have more than 2 random matches with a specific non-homologous sequence will be very low, won't it? So why do you have to count the number of k-mer matches at all instead of just demanding at least two matches?

This is due to multiple testing, which is now clear in the Methods subsection for phase 1.

* How does the k-mer anchoring similarity algorithm work? What makes it fast? How is the similarity defined? Total number of compatible k-mers? Or number per overlap?

This is now clear in the revised Methods.

* Line 652: "Second, another sequences is randomly selected with probability proportional to d_1 and inversely proportional to the non-overlapping difference in length with the first sequence." It is unclear what is calculated here. Please write down the probability explicitly as mathematical formula.

The probability formulas are now provided.

* Analysis for figures 2, 3, and 6: How were input set sizes reduced? By randomly sampling subsets of defined sizes?

This is now noted in the revised Methods.

* Are alignments between amino acid sequences computed using a standard substitution matrix?

The substitution matrix and alignment parameters are now provided.

2. Benchmarking.

a) Please include all TIGRFAM families in the input set. Explicitly excluding smaller families seems quite artificial.

The results on all TIGRFAM families are now shown. I decided to split this into two sets: 411 large and 3,001 small families, because on the small families it was possible to perform exact clustering. The large families are the same as in the previously benchmarked set, and the results are still shown in Fig. 2. The small families are shown in a new Figure S3. The new results on small families are nearly identical, corroborating the results on large families.

b) The only analysis of protein sequence clustering is shown Fig. 2, but it does not include an analysis of runtime and memory. Either inclusion in the analysis of Fig 3 or at least the runtimes on the full set for the 5 tools would be helpful, to see if Clusterize is competitive with the other tools.

The reviewer was correct to notice that runtime and memory analyses were previously only shown for nucleotide sequence sets. This has been addressed in the revised manuscript by addition of two different protein benchmarks. I think they tell a similar story to the nucleotide benchmarks, but now this information is included in the manuscript.

c) How does Clusterize perform with only the k-mer strategy in phase and with only the relatedness sorting in phase 3?

Clusterize keeps track of which method, relatedness sorting or rare k-mers, is the source of each sequence that is clustered. At the end of running, Clusterize outputs the fraction of clustered sequences originating from each method. To answer the reviewer's question, I performed an analysis of cluster origins on the large TIGRFAM protein families benchmark. As the reviewer can see in the new Figure S2, relatedness sorting is the origin of more clustered sequences at low similarity and the rare k-mer strategy is dominant at higher similarity thresholds. However, neither method is the exclusive cluster origin by a wide margin, i.e., less than 0.5% on average of clustered sequences come from only one method. There is substantial variance in which approach is preferred across different sequence sets, depending on the similarity threshold. Some input sequence sets cause Clusterize to draw more heavily from one method or the other, and it is clear both strategies are essential to the dependability of Clusterize for maintaining high accuracy across varied inputs.

d) Clusterize chooses 100 k-mers per sequence in phase 1 and 50 in phase 3. Linclust chooses by default only 20, and this can be changed by -m option. In their 2018 paper, linclust authors showed results for both -m20 and -m80 and -m80 had much better performance than the

default -m20 setting at only small runtime cost. So Linclust -m80 should be included in the Clusterize benchmarks (or replace the -m20 version).

I did exactly as the reviewer asked by replacing the 20 with 80 k-mers per sequence throughout. As the reviewer anticipated, sensitivity improved noticeably at a minor cost to speed. Nevertheless, Linclust's accuracy is still noticeably worse than Clusterize's accuracy, the degree to which depended on the benchmark.

e) I suspect the minimum coverage between sequences to be a strong hidden confounder in these benchmarks, as this setting is expected to have a much larger influence on the clustering quality than the sequence identity and the actual clustering algorithm. For instance, in Figure 6, at 100% sequence identity threshold, MMseqs and Linclust cluster much fewer sequences together than Clusterize and CDHIT, even though it is obviously trivial to find the 100% identical sequence pairs. The only explanation I can think of for this large divergence is different minimum coverage settings in the tools. Please check this and choose the same settings for all tools to ensure comparability across tools in all benchmarks.

I did as the reviewer asked and parameterized all algorithms to match coverage. I believe this resolved the issue the reviewer noted with some input sequence sets, and I thank the reviewer for identifying this issue. However, this change did not remedy MMseqs2's and Linclust's comparatively poor clustering quality on large sets of homologous sequences (Fig. 3), all of which already had very high coverage. I believe it is much more difficult to find highly similar sequences in very large sets of homologous sequences than diverse non-homologous sequences. It is only "obviously trivial to find the 100% identical pairs" if they are the same length, because hashing can be used with linear time complexity. Finding nearly identical sequences of different lengths is still challenging, as reflected by this being the slowest case for most programs.

3. What values are set for the parameters A,B,C,D,E set in the benchmarks? How were these values selected? Do any of these parameters depend on the input dataset? If yes, how?

I now show how default values were set for parameters A, B, C, and E. (Note, there is no parameter D to avoid confusion with the variable D described in the manuscript.) The sequences used to determine default parameter values were specifically chosen to be independent of the benchmarks shown in the manuscript, thereby separating training and test data. It consisted of 1.8 million diverse full-length non-coding RNA sequences with less than 800 nucleotides, which avoids the inclusion of 16S rRNA sequences (~1,500 nucleotides at full-length). The default parameters are fixed for all input sets, although they could be adjusted by the user to balance speed versus accuracy.

Reviewer #2 (Remarks to the Author):

I thank the author for the comments and additional work. Unfortunately, I could not say that my original concerns are dispelled with the revised version of the manuscript.

I appreciate the reviewer's feedback, which I believe improved the manuscript.

- I suggested using the very same dataset as the one used in the Linclust paper because I thought that this would be the best way of directly comparing approaches. Even if GO terms are not the best functional choice for benchmarking, such a comparison would have provided an orthogonal and straightforward validation of Clusterize, which I think would be one of the main questions that any potential user of Clusterize would like to answer: Can I cluster better and faster than Linclust/MMseqs? Even if ignoring GO terms, it would be possible to compare runtimes, number of clusters produced and other technical parameters.

In the revised version of the manuscript, I included two new benchmarks to address the reviewer's concern. In order to make space, I removed the old benchmarks that the reviewer did not find sufficient to address their original concerns. The first benchmark I added was Metaclust (Fig. 3), as the reviewer requested. Linclust and MMseqs2 were much faster than Clusterize on Metaclust. However, Linclust and MMseqs2 generated far more clusters than Clusterize at the same similarity threshold, even though they also clustered more sequences beyond the similarity threshold. This shows a clear trade-off between accuracy and speed for users to consider.

The second benchmark I added was designed to mirror the Linclust benchmark using Uniprot but include both nucleotide and protein sequences. I downloaded bacterial genomes from RefSeq, which are uniformly annotated with NCBI's PGAP annotation software. I assembled the set of all unique protein coding (nucleotide) and protein (amino acid) sequences from every genome that were not labeled "hypothetical protein", because this label provides no information for gauging accuracy. I then appended the sequences from randomly selected genomes to create a large, unbiased, and representative set of benchmark sequences with accurate annotations. I used the same strategy as Linclust, label consistency, to judge accuracy. This is very similar to Linclust's Uniprot benchmark, and I hope it assuages the reviewer's concern. Linclust and MMseqs2 were faster than Clusterize but produced clusters with lower label consistency (new Fig. 5).

- Dr. Wright argues that the new datasets added are enormous, but I do not agree. Linclust was tested on 1.6 billion metagenomic sequences. The newly added intergenic regions dataset is 32 million. So, the question remains: Is there any limitation on the size of the datasets that Clusterize can handle? I don't think that the performance and scalability increase claimed in the paper can be fully supported without a formal comparison based on datasets of at least a similar size range.

As stated above, larger benchmarks are now shown in the revised manuscript. The performance of Clusterize is shown out to 64 million protein sequences, where it reached the time limit. Linclust (and sometimes MMseqs2) were able to cluster far more sequences in the allotted time. It is clear in the manuscript that Clusterize and Linclust have similar (linear) scalability, although Linclust is faster and Clusterize is more accurate. Therefore, there is a clear speed

versus accuracy trade-off that is noted in the manuscript, and the user must make a choice as to which program better meets their needs. Linclust has a major time advantage for extremely large sequence sets, but Clusterize was far more accurate on some benchmarks. Time is the main limitation on the size of datasets Clusterize can handle, which in this manuscript was capped at 48 hours. Clusterize is designed to handle up to $2^{31} - 1$ (2,147,483,647) input sequences if a user has sufficient time.

- I also suggested improving/expanding the TIGRFAM benchmark, as it was very small (only 411 protein families) and perhaps biased due to various confounders. As an alternative, I suggested comparing Clusterize results with the exact clustering algorithms, just to confirm that results were accurate from a technical point of view. Unfortunately, none of the options were possible. Generating ground-truth results based on exact-clustering algorithms was argued to be impractical and therefore not addressed. However, I still think that, for such as a small set of sequences, other controls could have been used. 20,000 sequences are in principle addressable by hierarchical clustering algorithms or orthology clustering methods.

I appreciate the reviewer's suggestion here. As I mentioned in the first review, "most of the families have far more than 20k sequences and it is impractical to hierarchically cluster this many sequences." The largest protein family, with 108k sequences, is too large to hierarchically cluster with accuracy. However, the first reviewer asked for the benchmark to include all TIGRFAM families less than 20k sequences. For these sets it is practical to perform exact clustering, which I added to the updated manuscript (Fig. S3). I hope this addresses the reviewer's concern. Notably, exact hierarchical clustering showed better AMI and NMI than all inexact clustering programs (including Clusterize), confirming the merits of exact clustering and corroborating the use of AMI and NMI with taxonomy as proxies for accuracy.

- Also, the new panel in Figure 2 showing benchmarking results at the class level does not solve the potential biases. Actually, it seems to suggest that CD-HIT and MMSeqs are performing better (higher peaks) than Clusterize.

I think the reviewer meant that UCLUST (not CD-HIT) and MMseqs2 showed higher peaks than Clusterize in Figure 2. This is possibly true for UCLUST, although the case is less clear for MMseqs2, which has slightly higher AMI and slightly lower peak NMI. The difference between these top three programs is minimal, and there is no difference between Clusterize and UCLUST on the new set of 3,001 TIGRFAM families, although MMseqs2 has a slightly lower peak (Fig. S3). However, Linclust and CD-HIT have consistently lower peak AMI and NMI on both large and small TIGRFAM benchmarks (Figs. 2 and S3).

- Most importantly, even if the accuracy of all methods are considered very similar (as Figure 2 seems to suggest), the only substantial benefit of Clusterize would reside in runtimes and scalability improvements, which, as I mentioned before, I don't think it has been fully proven yet.

I completely disagree with the reviewer's point here. I do not believe Figure 2 shows the programs as have similar accuracy, and the other benchmarks do not support this assertion either. Linclust had substantially lower sensitivity than most other programs, and CD-HIT had notably lower sensitivity at low similarity thresholds on the TIGRFAM benchmarks (Figs. 2 and S3). Both UCLUST and CD-HIT typically displayed far worse scalability than Clusterize, as well as MMseqs2 at high similarity thresholds. Also, the SARS-CoV-2 and 16S analyses clearly showed that Clusterize generated far better clusters than Linclust or MMseqs2 – Clusterize's clusters were much larger, especially when accounting for the fraction of clustered sequences exceeding the user-specified similarity threshold. The new Metaclust and RefSeq benchmarks further confirm these results, demonstrating that Clusterize is more accurate than MMseqs2 and Linclust and more scalable and/or accurate than UCLUST and CD-HIT. For these reasons, I do not believe this comment reflects a fair interpretation of the results shown in any version of this manuscript. Indeed, Linclust (and sometimes MMseqs2) is faster than Clusterize, but this is at the expense of accuracy. I added the comparable number of output clusters in Fig. 3 to the Results text in order to reinforce this point. These numbers are summarized in the table below:

Sequences	Similarity threshold	Input sequences	Clusterize clusters	Linclust clusters
SARS-CoV-2	99.99%	2 million	957,945	1,919,893
16S reads	99.00%	16 million	1,954,730	9,050,721
Metaclust	50.00%	16 million	6,285,025	8,339,747

Both Linclust and Clusterize have similar (linear) scalability. Yet, the table shows that their number of output clusters differs by millions, a dramatic difference. Therefore, I do not think it is fair to say "the only substantial benefit of Clusterize would reside in runtimes and scalability improvements." The results show that no other program was simultaneously as accurate, scalable, and fast as Clusterize across the wide variety of benchmarks.

Reviewer #1 (Remarks to the Author):

The benchmarks, and the manuscript have been improved dramatically. The methods section now gives a readable account of the quite involved algorithm developed for Clusterize. Also, the algorithm's phase 1 has been redeveloped.

The results show Clusterize to be a good clustering tool that performs well on all tested benchmarks. Its greatest strength is clustering at very high sequence identity, as can be seen in the 99.99% clustering of Sars-Cov-2 genomes, and 16S rRNA sequences. In both, it performs a bit better than and similarly well as CD-HIT and ESPRIT-forest and better than the other programs. On Metaclust, it is much slower than Linclust and MMseqs but produces slightly fewer clusters.

Its main limitation is the size of the input dataset, which seems to be limited to somewhere around 50M sequences.

There are a few minor issues remaining:

1. The abstract states that Clusterize exhibits better scalability than MMseqs and UCLUST, but judging from the four benchmarks in Fig. 5, its runtime scales similar to UCLUST and worse than MMseqs. The only exception are the clustering at 99.99% and 99% of the two nucleotide sequence sets in Fig 3, in which MMseqs performs and scales badly, while UCLUST is comparable to Clusterize.

2. The description of phase 3 is still not clear to me. Why first run phase 1 and phase 2 and then, in phase 3, use the same algorithms that were used in phases 1 and 2 again? Why is phase 3 needed at all?

"The relatedness sorting strategy selects proximal sequences from the final rank ordering of sequences determined in phase 2." Do phase 1 and 2 within phase 3 run independently of each other or does phase 2 use as input the output of phase 1?

3. Page 570: "each k-mer is randomly projected into the hashing space, and the frequency of each hash bin is tabulated." Each hash bin contains a large number $x^{\{k/2\}}$ of unrelated k-mers. Doesn't that mean that according to the central limit theorem these frequencies should all be very similar? Would that invalidate the approach?

4. In the methods description, partition and group are for two different entities, clusters after phase 1 and 2, respectively, but this is not done consistently, which is confusing.

5. In the discussion, the author writes "percent identity is a weak proxy for homology at low levels of similarity, making defining a clustering threshold problematic." I agree. Perhaps this is the reason why mmseqs actually converts the sequence identity into an alignment score per aligned residue and uses this as clustering threshold, as explained in their documentation. This might explain why mmseqs produces clusters with more pairs of sequences above the sequence identity clustering threshold for low sequence identities. This might be worth mentioning in the context of figures 2 and 5.

Reviewer #2 (Remarks to the Author):

I appreciate the author's comments and clarifications in this new revision.

Also, I want to emphasize that my suggestions only aimed to reduce the ambiguity present in the paper. I am actually eager to have better clustering methods in our toolbox. However, it was difficult to judge when Clusterize is a better option compared to current methods.

I understand that the author disagrees with my interpretation of some data and figures, but this is probably due to the same ambiguity I am referring to. For instance, in the response to my last comment, it seems that the author agrees with my interpretation that Figure 2 shows that other methods might produce better accuracy scores than Clusterize.

When I said "(Figure 2 seems to suggest that UCLUST* and MMSeqs are performing better (higher peaks) than Clusterize", the author recognised that this is indeed a possible interpretation: "This is possibly true for UCLUST, although the case is less clear for MMseqs2, which has slightly higher AMI and slightly lower peak NMI".

However, when I said "even if the accuracy of all methods are considered very similar (as Figure 2 seems to suggest), the only substantial benefit of Clusterize would reside in runtimes and scalability improvements, which, as I mentioned before, I don't think it has been fully proven yet", the author disagreed arguing that "I do not believe Figure 2 shows the programs have similar accuracy, and the other benchmarks do not support this assertion either. Linclust had substantially lower sensitivity than most other programs, and CD-HIT had notably lower sensitivity at low similarity thresholds on the TIGRFAM benchmarks (Figs. 2 and S3)."

I completely agree that I should have omitted the word "all" in my sentence and I apologize for such imprecision, but please understand that the same concern remains if only UClust or MMSeqs2 are considered. For instance, if MMseqs provide similar accuracy and much better runtimes than Clusterize in this specific benchmark, what is the target niche of Clusterize?

It is true that other benchmarks, like the ones based on the number of clusters, show a better performance for Clusterize (and I actually think this is a good indication). But this is exactly what the AMI/NNI benchmarking section explicitly questioned in the manuscript, leading to contradicting interpretations:

"Clustering approaches are often compared by the number of clusters they generate at the same similarity threshold, although this comparison fails to take into account different definitions of similarity and linkage (e.g., average, complete, or single)".

Even the abstract is ambiguous in this regard. First, it states that "Clusterize produces clusters with accuracy rivaling popular programs (CDHIT, MMseqs2, and UCLUST) but exhibits better asymptotic scalability.", transmitting the idea that Clusterize has similar accuracy than other methods but it scales better (which is in line with the point I made in my last review). Then it states that "Clusterize generates higher accuracy and oftentimes much larger clusters than Linclust", suggesting higher accuracy. But, is the accuracy improvement only expected when compared with Linclust? If so, what's the advantage of Clusterize over MMseqs2?

My whole point here is that the value and target use-case of Clusterize is very difficult to delineate with the results shown.

My current interpretation is that, when datasets are not too big (i.e. a few million sequences) or are composed of highly-similar sequences, Clusterize is a very good option in terms of accuracy and speed (fewer clusters than other methods and affordable runtimes). However, for very large heterogeneous datasets such as protein databases or metagenomic catalogs, Clusterize would not be a feasible option.

Note that, already in the first review round, reviewer 1 and I asked about the possibility of clustering very large datasets (e.g. >1billion sequences), as this is a pressing need in many fields. The metaclust benchmark is now included in the manuscript, but for some reason the number of sequences clustered seems to be capped to 64-128M. In addition, the author states that "Clusterize is designed to handle

up to $2^{31} - 1$ (2,147,483,647) input sequences if a user has sufficient time.", which suggests that clustering very large datasets with Clusterize is theoretically possible, but hardly feasible in practice. This important limitation is now properly recognised in the discussion of the manuscript:

"For huge numbers of non-homologous sequences (> 100 million), MMseqs2 or Linclust remain the most practical option for clustering in a reasonable amount of time, although potentially at the expense of accuracy."

But this is not aligned with the abstract and other sections of the paper. For instance, the abstract introduces Clusterized from the perspective problem of clustering huge datasets:

"Search-based approaches to clustering scale super-linearly with the number of input sequences, making it impractical to cluster huge sets of sequences."

And the introduction correctly explains the challenge of exponential growth in databases:

"Exponential growth in the number of publicly available sequences, with a doubling time of approximately 3 years, will steadily increase the necessity of linear time clustering algorithms"

However, many reference databases and datasets are over 100M heterogeneous sequences, which I understand is not the type of problem where Clusterize is practical.

Overall, I really appreciate the value of this new method (even if it cannot handle huge datasets) but, if the manuscript is finally accepted, I think these limitations should be presented more clearly and without ambiguities.

I am very thankful to the reviewers for their constructive feedback throughout all versions of the manuscript. I especially appreciate the reviewers providing constructive feedback for multiple rounds of review. This revision clarifies the advantages and disadvantages of Clusterize in accordance with reviewer comments.

Reviewer #1 (Remarks to the Author):

The benchmarks, and the manuscript have been improved dramatically. The methods section now gives a readable account of the quite involved algorithm developed for Clusterize. Also, the algorithm's phase 1 has been redeveloped.

The results show Clusterize to be a good clustering tool that performs well on all tested benchmarks. Its greatest strength is clustering at very high sequence identity, as can be seen in the 99.99% clustering of Sars-Cov-2 genomes, and 16S rRNA sequences. In both, it performs a bit better than and similarly well as CD-HIT and ESPRIT-forest and better than the other programs. On Metaclust, it is much slower than Linclust and MMseqs but produces slightly fewer clusters.

I appreciate the reviewer's positive comments, and I agree that Clusterize performed particularly well when clustering at high sequence identity.

The reviewer's stated impression was that Clusterize offers a modest improvement over MMseqs2 and Linclust on Metaclust at low sequence identity. For example, clustering 16 million sequences yielded 6,285,025 clusters with Clusterize, 7,567,092 with MMseqs2, and 8,339,747 with Linclust. That is, MMseqs2 produced 1,282,067 (20%) more clusters than Clusterize, and Linclust produced 2,054,722 (33%) more clusters than Clusterize. This information is now included in the revised manuscript.

Its main limitation is the size of the input dataset, which seems to be limited to somewhere around 50M sequences.

I agree with the reviewer that runtime is a limitation for practical purposes. I will continue working on improving Clusterize's runtime in future versions.

There are a few minor issues remaining:

1. The abstract states that Clusterize exhibits better scalability than MMseqs and UCLUST, but judging from the four benchmarks in Fig. 5, its runtime scales similar to UCLUST and worse than MMseqs. The only exception are the clustering at 99.99% and 99% of the two nucleotide sequence sets in Fig 3, in which MMseqs performs and scales badly, while UCLUST is comparable to Clusterize.

To address this comment, I changed the Abstract from "exhibits better asymptotic scalability" to "exhibits linear asymptotic scalability".

I agree that the situation is more complex than can be quickly stated in the 150 word Abstract. Scalability of UCLUST and MMseqs2 appears to a function of the number of clusters and, therefore, both the similarity threshold, sequence diversity, and number of sequences. This is complex to convey precisely in the text. In the revised text, I have done my best to make it clear scalability is context dependent.

2. The description of phase 3 is still not clear to me. Why first run phase 1 and phase 2 and then, in phase 3, use the same algorithms that were used in phases 1 and 2 again? Why is phase 3 needed at all?

“The relatedness sorting strategy selects proximal sequences from the final rank ordering of sequences determined in phase 2.” Do phase 1 and 2 within phase 3 run independently of each other or does phase 2 use as input the output of phase 1?

I tried to clarify this point in the revised text. In summary, phases 1 and 2 are needed to determine the relatedness ordering, but no clustering is done until phase 3. Only a small subset of sequences are compared in phase 2, so clustering still needs to be performed in phase 3.

3. Page 570: “each k-mer is randomly projected into the hashing space, and the frequency of each hash bin is tabulated.” Each hash bin contains a large number $x^{\{k/2\}}$ of unrelated k-mers. Doesn't that mean that according to the central limit theorem these frequencies should all be very similar? Would that invalidate the approach?

If I understand the reviewer's point, the tabulation of rare k-mers is a sum of presence/absence variables, which would be expected to converge to a normal distribution due to central limit theorem. However, the existence of rare k-mers in the same sequence is highly correlated and, thus, strongly dependent. I believe we should anticipate (and do observe) an exponentially decreasing background analogous to the tail of a normal distribution. The sequences that contain multiple correlated k-mers should appear above this background, as they violate the independence (or weak dependence) assumption underlying the central limit theorem. Thus, no changes have been made to the text in response to this comment.

4. In the methods description, partition and group are for two different entities, clusters after phase 1 and 2, respectively, but this is not done consistently, which is confusing.

This is fixed in the revised manuscript.

5. In the discussion, the author writes “percent identity is a weak proxy for homology at low levels of similarity, making defining a clustering threshold problematic.” I agree. Perhaps this is the reason why mmseqs actually converts the sequence identity into an alignment score per aligned residue and uses this as clustering threshold, as explained in their documentation. This might explain why mmseqs produces clusters with more pairs of sequences above the sequence identity clustering threshold for low sequence identities. This might be worth mentioning in the context of figures 2 and 5.

I agree that this is a possible explanation, although I was able to find very little about this in the MMseqs2 documentation:

“[...] the equivalent similarity score of the local alignment (including gap penalties) divided by the maximum of the lengths of the two locally aligned sequence segments. The score per residue equivalent to a certain sequence identity is obtained by a linear regression using thousands of local alignments as training set.”

Nonetheless, I have added this possibility to the main text. It was unclear whether this scoring method applied to both MMseqs2 and Linclust or only MMseqs2.

Reviewer #2 (Remarks to the Author):

I appreciate the author's comments and clarifications in this new revision.

Also, I want to emphasize that my suggestions only aimed to reduce the ambiguity present in the paper. I am actually eager to have better clustering methods in our toolbox. However, it was difficult to judge when Clusterize is a better option compared to current methods.

I understand that the author disagrees with my interpretation of some data and figures, but this is probably due to the same ambiguity I am referring to. For instance, in the response to my last comment, it seems that the author agrees with my interpretation that Figure 2 shows that other methods might produce better accuracy scores than Clusterize.

When I said "(Figure 2 seems to suggest that UCLUST* and MMSeqs are performing better (higher peaks) than Clusterize)", the author recognised that this is indeed a possible interpretation: "This is possibly true for UCLUST, although the case is less clear for MMseqs2, which has slightly higher AMI and slightly lower peak NMI".

However, when I said "even if the accuracy of all methods are considered very similar (as Figure 2 seems to suggest), the only substantial benefit of Clusterize would reside in runtimes and scalability improvements, which, as I mentioned before, I don't think it has been fully proven yet", the author disagreed arguing that "I do not believe Figure 2 shows the programs have similar accuracy, and the other benchmarks do not support this assertion either. Linclust had substantially lower sensitivity than most other programs, and CD-HIT had notably lower sensitivity at low similarity thresholds on the TIGRFAM benchmarks (Figs. 2 and S3)."

I completely agree that I should have omitted the word "all" in my sentence and I apologize for such imprecision, but please understand that the same concern remains if only UClust or MMSeqs2 are considered. For instance, if MMseqs provide similar accuracy and much better runtimes than Clusterize in this specific benchmark, what is the target niche of Clusterize?

I thank the reviewer for clarifying their original point. I would argue that the target niche for Clusterize is dependability. Clusterize is consistently linear in time, consistently clusters a moderate number of sequences above the threshold, and consistently forms relatively few clusters. All other programs struggle in one or more of these dimensions depending on the benchmark. Admittedly, Clusterize struggles in scaling to hundreds of millions of sequences in practical times (e.g., a day). As I already pointed out in the Discussion, there is no panacea. I would qualitatively summarize the results as follows:

	TIGRFAM		SARS		16S		Metaclust		Genes - coding sequences		Genes - protein sequences	
	Accuracy	Obeying threshold	Accuracy	Speed	Accuracy	Speed	Accuracy	Speed	Accuracy	Speed	Accuracy	Speed
Clusterize	+++	++	+++	+++	+++	++	+++	+	++	++	++	++
CD-HIT	++	+	+++	++	+++	++	+++	-	++	+	++	+
Linclust	++	+	+	+++	+	+++	+	+++	+	+++	+	+++
MMseqs2	+++	+	+	-	+	+	++	++	+	+++	+	+++
UCLUST	+++	+++	-	+++	+	+	+	+	+++	+	+++	++

+++ = Good; ++ = Okay; + = Mediocre; - = Bad

Of course, this is my interpretation of the results and such an ambiguous rating scheme is unreliable. Nevertheless, my point is that Clusterize consistently scores well across most benchmarks, especially when holistically accounting for the number of clusters relative to the percent meeting threshold. Which program to pick really depends on the application, but Clusterize works well in most cases. Where it falls short of MMseqs2 is in quickly clustering large numbers of protein sequences at low similarity.

It is true that other benchmarks, like the ones based on the number of clusters, show a better performance for Clusterize (and I actually think this is a good indication). But this is exactly what the AMI/NNI benchmarking section explicitly questioned in the manuscript, leading to contradicting interpretations:

"Clustering approaches are often compared by the number of clusters they generate at the same similarity threshold, although this comparison fails to take into account different definitions of similarity and linkage (e.g., average, complete, or single)".

Even the abstract is ambiguous in this regard. First, it states that "Clusterize produces clusters with accuracy rivaling popular programs (CDHIT, MMseqs2, and UCLUST) but exhibits better asymptotic scalability.", transmitting the idea that Clusterize has similar accuracy than other methods but it scales better (which is in line with the point I made in my last review). Then it states that "Clusterize generates higher accuracy and oftentimes much larger clusters than Linclust", suggesting higher accuracy. But, is the

accuracy improvement only expected when compared with Linclust? If so, what's the advantage of Clusterize over MMseqs2?

I understand the reviewer's concern that the picture of which program to rely upon is not clear cut. Note the Abstract sentence quoted here Reviewer 2 was revised in the updated manuscript in accordance with Reviewer 1 feedback.

The TIGRFAM benchmark is limited in that it only contains homologous sequences, and for this reason it is necessary to consider the other benchmarks to gauge large-scale accuracy. Then, which program is highest accuracy is context dependent. I would argue that Clusterize was more accurate than Linclust and MMseqs2 on all benchmarks to different degrees, but especially for nucleotide sequences. By default, Clusterize is balanced toward accuracy over speed. I clarified this point in the Discussion section.

My whole point here is that the value and target use-case of Clusterize is very difficult to delineate with the results shown.

I think it comes down to the user preferences and practical constraints, which makes it impossible to declare a single program as best. This was the situation before Clusterize, and it remains the situation except that Clusterize offers a dependable option that will work well in many cases. The choice of accuracy versus speed partly comes down to personal preference and I do not think there can be a single answer for everyone.

My current interpretation is that, when datasets are not too big (i.e. a few million sequences) or are composed of highly-similar sequences, Clusterize is a very good option in terms of accuracy and speed (fewer clusters than other methods and affordable runtimes). However, for very large heterogeneous datasets such as protein databases or metagenomic catalogs, Clusterize would not be a feasible option.

We are in agreement. I will continue to work on Clusterize's speed in future versions.

Note that, already in the first review round, reviewer 1 and I asked about the possibility of clustering very large datasets (e.g. >1billion sequences), as this is a pressing need in many fields. The metaclust benchmark is now included in the manuscript, but for some reason the number of sequences clustered seems to be capped to 64-128M.

The Metaclust benchmark was capped at 256 million sequences because of memory limitations. Nevertheless, I think it sufficiently made the point that Linclust can easily cluster 256 million sequences in less than half a day given a computer with sufficient memory. I did not focus on the cap in the text because it is clear that Linclust outcompetes all other programs on this benchmark, and this finding was previously shown in the Linclust paper.

In addition, the author states that "Clusterize is designed to handle up to $2^{31} - 1$ (2,147,483,647) input sequences if a user has sufficient time.", which suggests that clustering very large datasets with Clusterize is theoretically possible, but hardly

feasible in practice. This important limitation is now properly recognised in the discussion of the manuscript:

"For huge numbers of non-homologous sequences (> 100 million), MMseqs2 or Linclust remain the most practical option for clustering in a reasonable amount of time, although potentially at the expense of accuracy."

But this is not aligned with the abstract and other sections of the paper. For instance, the abstract introduces Clusterize from the perspective problem of clustering huge datasets:

"Search-based approaches to clustering scale super-linearly with the number of input sequences, making it impractical to cluster huge sets of sequences."

And the introduction correctly explains the challenge of exponential growth in databases:

"Exponential growth in the number of publicly available sequences, with a doubling time of approximately 3 years, will steadily increase the necessity of linear time clustering algorithms"

However, many reference databases and datasets are over 100M heterogeneous sequences, which I understand is not the type of problem where Clusterize is practical.

This comes down to the definition of huge. Sets of a billion might be huge now but not in the future. I changed the word "huge" to "very large" in the Abstract to address this comment. I agree with the reviewer that, at present, Clusterize would be impractical for clustering billions of sequences.

Overall, I really appreciate the value of this new method (even if it cannot handle huge datasets) but, if the manuscript is finally accepted, I think these limitations should be presented more clearly and without ambiguities.

I refined the revised text to mitigate any ambiguities about the limitations of Clusterize. I thank the reviewer again for the constructive review.

Reviewer #1 (Remarks to the Author):

The issues I raised have been addressed appropriately. I congratulate the author on the nice work.

Reviewer #2 (Remarks to the Author):

Thanks for the clarifications. The pros and cons of the method are now more clearly disclosed in the paper.

Reviewer #2 (Remarks on code availability):

I have only tested some examples and check the benchmark code to understand some references, but haven't looked into the code of the software itself, nor re-execute the benchmark scripts.